# Stochastic Optimization Schemes for Performative Prediction with Nonconvex Loss

**Qiang Li**      **Hoi-To Wai**

Department of Systems Engineering and Engineering Management
The Chinese University of Hong Kong, Shatin, Hong Kong SAR of China
`{liqiang, htwai}@se.cuhk.edu.hk`

## Abstract

This paper studies a risk minimization problem with decision dependent data distribution. The problem pertains to the performative prediction setting in which a trained model can affect the outcome estimated by the model. Such dependency creates a feedback loop that influences the stability of optimization algorithms such as stochastic gradient descent (SGD). We present the first study on performative prediction with smooth but possibly non-convex loss. We analyze a greedy deployment scheme with SGD (SGD-GD). Note that in the literature, SGD-GD is often studied with strongly convex loss. We first propose the definition of stationary performative stable (SPS) solutions through relaxing the popular performative stable condition. We then prove that SGD-GD converges to a biased SPS solution in expectation. We consider two conditions of sensitivity on the distribution shifts: (i) the sensitivity is characterized by Wasserstein-1 distance and the loss is Lipschitz w.r.t. data samples, or (ii) the sensitivity is characterized by total variation (TV) divergence and the loss is bounded. In both conditions, the bias levels are proportional to the stochastic gradient's variance and sensitivity level. Our analysis is extended to a lazy deployment scheme where models are deployed once per several SGD updates, and we show that it converges to an SPS solution with reduced bias. Numerical experiments corroborate our theories.

## 1 Introduction

When trained models are deployed in social contexts, the outcomes these models aim to predict can be influenced by the models themselves. Taking email spam detection as an example. On one hand, email service providers design filters to protect their users by identifying spam emails. On the other hand, spammers aim to circumvent these filters to distribute malware and advertisements. Each time a new classifier is deployed, spammers who are interspersed within the general population may alter the characteristics of their messages to evade detection. The above example pertains to the strategic classification problem [Dalvi et al., 2004, Cai et al., 2015, Hardt et al., 2016, Björkegren et al., 2020] and can be modelled by dataset shifts [Quiñonero-Candela et al., 2022].

The scenarios described can be captured by the recently proposed performative prediction problem, which called the above dataset shift phenomena as the 'performative' effect. Perdomo et al. [2020] proposed to study the risk minimization problem with a *decision-dependent* data distribution:

$$\min_{\boldsymbol{\theta} \in \mathbb{R}^d} \ V(\boldsymbol{\theta}) := \mathbb{E}_{Z \sim \mathcal{D}(\boldsymbol{\theta})}[\ell(\boldsymbol{\theta}; Z)], \tag{1}$$

where $\ell(\boldsymbol{\theta}; z)$ is a loss function that is continuously differentiable with respect to (w.r.t.) $\boldsymbol{\theta}$ for any given data sample $z \in \mathsf{Z}$, and $\mathsf{Z} \subseteq \mathbb{R}^p$ is the sample space. The dependence on $\boldsymbol{\theta}$ in $\mathcal{D}(\boldsymbol{\theta})$ explicitly captures the distribution shift effect of prediction models on data samples. The objective function $V(\boldsymbol{\theta})$ is also known as the performative risk.

38th Conference on Neural Information Processing Systems (NeurIPS 2024).

| Literature | Ncvx-$\ell$ | Ncvx-$V$ | Sensitivity | Algorithm | Rate | $\theta_\infty$ |
|---|---|---|---|---|---|---|
| [Izzo et al., 2021] | ✗ | ✓ | Loc.$^\dagger$ | 2-Phase | $\mathcal{O}(T^{-\frac{1}{2}})$ | $\nabla V(\cdot) = \mathbf{0}$ |
| [Miller et al., 2021] | ✗ | ✗ | $W_1$ | 2-Phase | $\mathcal{O}(T^{-1})$ | $\min V(\boldsymbol{\theta})$ |
| [Mendler-Dünner et al., 2020] | ✗ | ✓ | $W_1$ | SGD-GD | $\mathcal{O}(T^{-1})$ | PS |
| [Mofakhami et al., 2023] | ✗$^\ddagger$ | ✓ | $\chi^2$ | RRM$^\ddagger$ | Linear$^\ddagger$ | PS$^\ddagger$ |
| **This Work** | ✓ | ✓ | TV or $W_1$ | SGD-GD | $\mathcal{O}(T^{-\frac{1}{2}})$ | $\mathcal{O}(\epsilon)$-SPS |
|  | ✓ | ✓ | TV or $W_1$ | SGD-Lazy$^\star$ | $\mathcal{O}(T^{-\frac{1}{2}})$ | $\mathcal{O}(\epsilon^2)$-SPS |

Table 1: **Comparison of Results in Existing Works**. 'Sensitivity' indicates the distance metric imposed on $\mathcal{D}(\boldsymbol{\theta})$ when the latter is subject to perturbation, given in the form $d(\mathcal{D}(\boldsymbol{\theta}), \mathcal{D}(\boldsymbol{\theta}')) \leq \epsilon \|\boldsymbol{\theta} - \boldsymbol{\theta}'\|$ such that $d(\cdot, \cdot)$ is a distance metric between distributions. '$\theta_\infty$' indicates the type of convergent points: 'PS' refers to performative stable solution [cf. (4)], 'SPS' refers to Def. 1.

$^\dagger$Izzo et al. [2021] assumed that $\mathcal{D}(\boldsymbol{\theta})$ belongs to the location family, i.e., $\mathcal{D}(\boldsymbol{\theta}) = \mathcal{N}(f(\boldsymbol{\theta}); \sigma^2)$.

$^\ddagger$Mofakhami et al. [2023] considered $\ell(\boldsymbol{\theta}; z) = \tilde{\ell}(f_{\boldsymbol{\theta}}(x), y)$ with strongly convex $\tilde{\ell}(\cdot, y)$. The RRM requires solving a non-convex optimization at each recursion.

$^\star$SGD-Lazy refers to the SGD method with lazy deployment scheme, which fixes the deployed model for $K$ iterations before the next deployment; see §4.

The decision variable $\boldsymbol{\theta}$ in (1) affects simultaneously the distribution and the loss function. As such, optimizing $V(\boldsymbol{\theta})$ directly is often difficult. Mendler-Dünner et al. [2020] considered the following stochastic gradient (SGD) recursion: for any $t \geq 0$ and let $\gamma_{t+1} > 0$ be a stepsize,

$$\boldsymbol{\theta}_{t+1} = \boldsymbol{\theta}_t - \gamma_{t+1} \nabla \ell(\boldsymbol{\theta}_t; Z_{t+1}), \text{ where } Z_{t+1} \sim \mathcal{D}(\boldsymbol{\theta}_t). \tag{2}$$

The above is known as the *greedy deployment* scheme with SGD (SGD-GD), where the learner deploys the current trained model $\boldsymbol{\theta}_t$ before drawing samples from $\mathcal{D}(\boldsymbol{\theta}_t)$. The SGD-GD scheme describes a training procedure when the learner is *unaware of the performative phenomena* with the data distribution $\mathcal{D}(\cdot)$, which is plausible in many applications. Relevant studies to (2) include lazy deployment where the learner deploys a new model only once every few iterations, or repeated risk minimization; see [Mendler-Dünner et al., 2020, Perdomo et al., 2020, Zrnic et al., 2021].

Existing convergence analysis of (2) are limited to the case when $\ell(\boldsymbol{\theta}; z)$ is *strongly convex*[1] w.r.t. $\boldsymbol{\theta}$. Perdomo et al. [2020] introduced the concept of *performative stable* (PS) solution as the unique minimizer of (1) with fixed distribution. The PS solution, while being different from an optimal or stationary solution to (1), is shown to be the unique limit point of the recursion (2) provided that the sensitivity of the distribution map $\mathcal{D}(\cdot)$, measured w.r.t. the Wasserstein-1 ($W_1$) distance, is upper bounded by a factor proportional to the strong convexity modulus of $\ell(\boldsymbol{\theta}; z)$ [Mendler-Dünner et al., 2020]. Furthermore, such convergence condition is proven to be tight [Perdomo et al., 2020] and the analysis has been extended to proximal algorithm [Drusvyatskiy and Xiao, 2023], online optimization [Cutler et al., 2023], saddle point seeking [Wood and Dall'Anese, 2023], multi-agent consensus learning [Li et al., 2022], non-cooperative learning [Wang et al., 2023, Narang et al., 2023, Piliouras and Yu, 2023], state-dependent learning [Brown et al., 2022, Li and Wai, 2022], etc.

This paper provides the *first analysis* of SGD-GD and related stochastic optimization schemes in performative prediction when $\ell(\boldsymbol{\theta}; z)$ is *smooth but possibly non-convex*. This is a more common scenario in machine learning than the strongly convex loss considered in the prior works, e.g., it covers the case of training neural network (NN) models. We notice that existing works are limited to imposing structure on the loss function $\ell(\boldsymbol{\theta}; z)$ and the distribution $\mathcal{D}(\boldsymbol{\theta})$, utilizing advanced algorithms that demand extra knowledge on $\mathcal{D}(\boldsymbol{\theta})$, etc., as we overview below.

**Related Works.** In the non-convex setting, the most related work to ours is [Mofakhami et al., 2023] which proved that a variant of PS solution can be found when training NN in the performative prediction setting, i.e., a special case with non-convex loss. However, their analysis is restrictive: (i) it requires a loss function of the form $\ell(\boldsymbol{\theta}; z) = \tilde{\ell}(f_{\boldsymbol{\theta}}(x); y)$ where $\tilde{\ell}(\hat{y}; y)$ is strongly convex w.r.t. $\hat{y}$, (ii) it only analyzes the case of training NN using a repeated risk minimization (RRM) procedure which *exactly* minimizes a non-convex objective function at each step. In comparison, we concentrate on stochastic (first order) optimization schemes and require only smoothness for $\ell(\cdot; z)$.

---

[1]Note that $V(\boldsymbol{\theta})$ is still non-convex.

Other works departed from tackling the PS solution and considered alternative algorithms to directly minimize $V(\boldsymbol{\theta})$. For example, Roy et al. [2022] assumed that unbiased estimates of $\nabla V(\boldsymbol{\theta})$ is available and studied the convergence of stochastic conditional gradient algorithms towards a stationary solution of $V(\boldsymbol{\theta})$, Li and Wai [2022] assumed bounded biasedness w.r.t. $\nabla V(\boldsymbol{\theta})$ in (2) and show that (2) converges to a biased stationary point of $V(\boldsymbol{\theta})$. Notice that estimating $\nabla V(\boldsymbol{\theta})$ requires knowledge on $\mathcal{D}(\boldsymbol{\theta})$ which has to be learnt separately. To circumvent this difficulty, two phases algorithms are studied in [Miller et al., 2021, Izzo et al., 2021] that learn $\mathcal{D}(\boldsymbol{\theta})$ via a large batch of samples at the first stage, then optimize $\boldsymbol{\theta}$ later (see [Zhu et al., 2023] for a two-timescale type online algorithm), derivative free optimization are studied in [Miller et al., 2021, Ray et al., 2022, Liu et al., 2023], and confidence bound methods in [Jagadeesan et al., 2022]. In addition, Miller et al. [2021] proposed a mixture dominance assumption that can imply the strong convexity of $V(\boldsymbol{\theta})$. We remark that [Zhao, 2022] studied conditions to ensure $V(\boldsymbol{\theta})$ to be weakly convex. Our work does not require such advanced algorithms and show that stochastic (first order) optimization converges towards a similar solution as the PS solution. We display a comparison between these related works in Table 1.

**Our Contributions**: This work provides the *first* analysis of stochastic gradient-based methods for performative prediction with smooth but possibly *non-convex* losses. Our contributions are:

- We propose the concept of *stationary performative stable* (SPS) solutions which is a relaxation of the commonly used performative stable (PS) condition [Perdomo et al., 2020]. The relaxation is necessary for handling non-convex losses using first-order methods.

- We show that the stochastic gradient method with greedy deployment (SGD-GD) finds a biased SPS solution. Assume that the distribution shift is $\epsilon$-sensitive, i.e., it holds $d(\mathcal{D}(\boldsymbol{\theta}), \mathcal{D}(\boldsymbol{\theta}')) \leq \epsilon \|\boldsymbol{\theta} - \boldsymbol{\theta}'\|$ for some distance measure $d(\cdot, \cdot)$ between the shifted distributions $\mathcal{D}(\boldsymbol{\theta}), \mathcal{D}(\boldsymbol{\theta}')$, SGD-GD converges at a rate of $\mathcal{O}(T^{-\frac{1}{2}})$ in expectation to an $\mathcal{O}(\epsilon)$-SPS solution. The bias level is further improved to $\mathcal{O}(\epsilon^2)$ when the gradient is exact.

- Our analysis relies on constructing a time varying Lyapunov function that may shed new lights for non-gradient stochastic approximation [Dieuleveut et al., 2023]. We studied two alternative conditions on the distance metric between distributions. When $d(\cdot, \cdot)$ is the Wasserstein-1 distance, SGD-GD converges to a biased SPS solution for Lipschitz loss function. When $d(\cdot, \cdot)$ is the total variation (TV) distance, SGD-GD converges to a biased SPS solution for bounded loss function.

- We extend the analysis to the lazy deployment scheme with SGD [Mendler-Dünner et al., 2020]. The latter scheme finds an *SPS solution with reduced bias* as the epoch length of lazy deployment grows.

Lastly, we provide numerical examples on synthetic and real data to validate our theoretical findings. The rest of this paper is organized as follows. §2 introduces the problem setup and assumptions for establishing our convergence results of SGD-GD. Furthermore, we highlight the challenges in analyzing non-convex performative prediction. §3 introduces the concept of SPS solutions and presents the convergence results for SGD-GD under two alternative assumptions on the distribution shifts. We also outline the use of a time varying Lyapunov function to handle the dynamic nature of SGD-GD. §4 shows the results for the lazy deployment scheme. Lastly, §5 provides numerical experiments to illustrate our results.

**Notations.** Let $\mathbb{R}^d$ be the $d$-dimensional Euclidean space equipped with inner product $\langle \cdot \,|\, \cdot \rangle$ and induced norm $\|x\| = \sqrt{\langle x \,|\, x \rangle}$. Let $\mathcal{S}$ be a (measurable) sample space, and $\mu, \nu$ are two probability measures defined as $\mathcal{S}$. $\mathbb{E}[\cdot]$ denotes taking expectation w.r.t all randomness, $\mathbb{E}_t[\cdot] := \mathbb{E}_t[\cdot|\mathcal{F}_t]$ means taking conditional expectation on the filtration $\mathcal{F}_t := \sigma(\{\boldsymbol{\theta}_0, \boldsymbol{\theta}_1, \cdots, \boldsymbol{\theta}_t\})$, where $\sigma(\cdot)$ is the sigma-algebra generated by the random variables in the operand and $\{\boldsymbol{\theta}_t\}$ is the sequence of iterates generated by the SGD-GD scheme (2).

## 2 Stationary Condition for Performative Stability

This section prepares the analysis of (1) with SGD-GD and related schemes in the non-convex loss setting. To fix idea, we define the decoupled performative risk and the decoupled partial gradient:

$$J(\boldsymbol{\theta}_1; \boldsymbol{\theta}_2) = \mathbb{E}_{Z \sim \mathcal{D}(\boldsymbol{\theta}_2)}\left[\ell(\boldsymbol{\theta}_1; Z)\right], \quad \nabla J(\boldsymbol{\theta}_1; \boldsymbol{\theta}_2) = \mathbb{E}_{Z \sim \mathcal{D}(\boldsymbol{\theta}_2)}\left[\nabla \ell(\boldsymbol{\theta}_1; Z)\right]. \tag{3}$$

Observe that while $V(\boldsymbol{\theta}) = J(\boldsymbol{\theta}; \boldsymbol{\theta})$, $\nabla J(\boldsymbol{\theta}; \boldsymbol{\theta}) \neq \nabla V(\boldsymbol{\theta})$ in general since $\nabla J(\boldsymbol{\theta}; \boldsymbol{\theta})$ only represents a partial gradient of $V(\boldsymbol{\theta})$; see [Izzo et al., 2021]. In (2), the conditional expectation of the stochastic gradient update term satisfies $\mathbb{E}_t[\nabla\ell(\boldsymbol{\theta}_t; Z_{t+1})] = \nabla J(\boldsymbol{\theta}_t; \boldsymbol{\theta}_t)$.

If the loss $\ell(\boldsymbol{\theta}; z)$ is strongly convex w.r.t. $\boldsymbol{\theta}$, then the decoupled performative risk $J(\boldsymbol{\theta}; \bar{\boldsymbol{\theta}})$ admits a unique minimizer w.r.t. $\boldsymbol{\theta}$ for any $\bar{\boldsymbol{\theta}}$. It has hence motivated [Perdomo et al., 2020] to study the *performative stable* (PS) solution $\boldsymbol{\theta}_{PS}$ which is defined as a fixed point to the map

$$\mathcal{T}(\bar{\boldsymbol{\theta}}) := \arg\min_{\boldsymbol{\theta} \in \mathbb{R}^d} J(\boldsymbol{\theta}; \bar{\boldsymbol{\theta}}), \ \ \text{i.e., } \boldsymbol{\theta}_{PS} = \mathcal{T}(\boldsymbol{\theta}_{PS}). \tag{4}$$

In the above, the uniqueness and existence of $\boldsymbol{\theta}_{PS}$ follows by observing that $\mathcal{T}(\cdot)$ is a contraction if and only if the sensitivity of $\mathcal{D}(\boldsymbol{\theta})$, i.e., the 'smoothness' of $\mathcal{D}(\boldsymbol{\theta})$ w.r.t. $\boldsymbol{\theta}$, is upper bounded by the inverse of condition number of $\ell(\boldsymbol{\theta}; z)$. The convergence of SGD-GD follows by analyzing (2) as a stochastic approximation (SA) scheme for the repeated risk minimization (RRM) procedure $\boldsymbol{\theta} \leftarrow \mathcal{T}(\boldsymbol{\theta})$ [Mendler-Dünner et al., 2020].

For the case of *non-convex* loss in this paper, the analysis becomes more nuanced since the map $\mathcal{T}(\cdot)$ is no longer well-defined, e.g., there may exist more than one minimizers to $\min_{\boldsymbol{\theta}} J(\boldsymbol{\theta}; \bar{\boldsymbol{\theta}})$. Our remedy is to concentrate on the following non-convex counter part to the PS solution:

**Definition 1.** *($\delta$ stationary performative stable solution) Let $\delta \geq 0$, the vector $\boldsymbol{\theta}_{\delta-SPS} \in \mathbb{R}^d$ is said to be an $\delta$ stationary performative stable ($\delta$-SPS) solution of (1) if:*

$$\|\nabla J(\boldsymbol{\theta}_{\delta-SPS}; \boldsymbol{\theta}_{\delta-SPS})\|^2 = \left\|\mathbb{E}_{Z \sim \mathcal{D}(\boldsymbol{\theta}_{\delta-SPS})}[\nabla\ell(\boldsymbol{\theta}_{\delta-SPS}; Z)]\right\|^2 \leq \delta. \tag{5}$$

We also say that $\boldsymbol{\theta}_{SPS}$ is an (exact) SPS solution if it satisfies (5) with $\delta = 0$. In other words, $\delta \geq 0$ measures the stationarity of a solution. Notice that if $\ell(\boldsymbol{\theta}; z)$ is strongly convex w.r.t. $\boldsymbol{\theta}$, then an SPS solution is also a PS solution defined in [Perdomo et al., 2020].

Although (5) is similar to the usual definitions of stationary solution in smooth optimization, there is a subtle but critical difference since $\nabla J(\boldsymbol{\theta}; \boldsymbol{\theta})$ may not be the *gradient* of any function in $\boldsymbol{\theta}$. For example, consider $\ell(\boldsymbol{\theta}; z) = (1/2)\|\boldsymbol{\theta} - z\|^2$ and $\mathcal{D}(\boldsymbol{\theta}) \equiv \mathcal{N}(A\boldsymbol{\theta}, \boldsymbol{I})$ for some square but asymmetric matrix $A$, the map $\nabla J(\boldsymbol{\theta}; \boldsymbol{\theta}) = (I - A)\boldsymbol{\theta}$ has a Jacobian of $I - A$ which is not symmetric. Furthermore, we observe that the mean field for SGD-GD scheme (2) is $\nabla J(\boldsymbol{\theta}; \boldsymbol{\theta})$, which is not a gradient. The SGD-GD scheme is thus a special case of non-gradient SA scheme [Dieuleveut et al., 2023].

To get further insight, as investigated in [Dieuleveut et al., 2023], a common analysis framework of non-gradient SA scheme is by identifying a smooth Lyapunov function linked to the recursion (2). When $\ell(\cdot; z)$ is strongly convex, we may study the Lyapunov functions as the squared distance $\|\boldsymbol{\theta} - \boldsymbol{\theta}_{PS}\|^2$ [cf. (4)]. It can be shown that the properties required in [Dieuleveut et al., 2023] are satisfied under the conditions analyzed by [Mendler-Dünner et al., 2020]. However, in the case of non-convex loss, identifying a suitable Lyapunov function for non-gradient SA scheme is hard. In the next section, we demonstrate how to address this challenge by identifying a time varying Lyapunov function for SGD-GD.

## 3  Main Results

This section presents theoretical results on the SGD-GD scheme with non-convex loss. We first show how to construct a time varying Lyapunov function tailor made for (2). We then show the convergence of SGD-GD under two different sets of conditions.

We introduce two basic and natural assumptions on the risk minimization problem (1):

**A1.** *For any $z \in \mathsf{Z}$, there exists a constant $L \geq 0$ such that*

$$\|\nabla\ell(\boldsymbol{\theta}; z) - \nabla\ell(\boldsymbol{\theta}'; z)\| \leq L\|\boldsymbol{\theta} - \boldsymbol{\theta}'\|, \ \forall \boldsymbol{\theta}, \boldsymbol{\theta}' \in \mathbb{R}^d, \tag{6}$$

*where $\nabla\ell(\boldsymbol{\theta}; z)$ denotes the gradient of $\ell(\boldsymbol{\theta}; z)$ w.r.t. $\boldsymbol{\theta}$. Moreover, there exists a constant $\ell^\star > -\infty$ such that $\ell(\boldsymbol{\theta}; z) \geq \ell^\star$ for any $\boldsymbol{\theta} \in \mathbb{R}^d$.*

**A2.** *For any fixed $\boldsymbol{\theta}_1, \boldsymbol{\theta}_2 \in \mathbb{R}^d$, the stochastic gradient $\nabla\ell(\boldsymbol{\theta}_1; Z)$, $Z \sim \mathcal{D}(\boldsymbol{\theta}_2)$ is unbiased such that $\mathbb{E}_{Z \sim \mathcal{D}(\boldsymbol{\theta}_2)}[\nabla\ell(\boldsymbol{\theta}_1; Z)] = \nabla J(\boldsymbol{\theta}_1; \boldsymbol{\theta}_2)$, and there exists constants $\sigma_0, \sigma_1 \geq 0$ such that*

$$\mathbb{E}_{Z \sim \mathcal{D}(\boldsymbol{\theta}_2)}\left[\|\nabla\ell(\boldsymbol{\theta}_1; Z) - \nabla J(\boldsymbol{\theta}_1; \boldsymbol{\theta}_2)\|^2\right] \leq \sigma_0^2 + \sigma_1^2\|\nabla J(\boldsymbol{\theta}_1; \boldsymbol{\theta}_2)\|^2. \tag{7}$$

Note that A1, 2 are standard assumptions that hold for a wide range of applications and the respective stochastic optimization based training schemes. For instance, A1 requires the loss function to be smooth, while A2 assumes the stochastic gradient estimates to have a variance that may grow with $\|\nabla J(\boldsymbol{\theta}_1; \boldsymbol{\theta}_2)\|^2$. For the *non performative prediction* setting where the data distribution is not shifted by $\boldsymbol{\theta}$, i.e., $\mathcal{D}(\boldsymbol{\theta}) \equiv \mathcal{D}$, these assumptions guarantee that SGD algorithm (i.e., SGD-GD with $Z_{t+1} \sim \mathcal{D}$) to converge to a stationary solution of (1) with a suitable step size schedule [Ghadimi and Lan, 2013]. In particular, in this case A1 implies that $\mathbb{E}_{Z \sim \mathcal{D}}[\ell(\boldsymbol{\theta}; Z)]$ is a smooth function and serves as a Lyapunov function for the SGD algorithm.

In this light, it might be tempting to use the performative risk $V(\boldsymbol{\theta})$ [cf. (1)] as the Lyapunov function for (2) and directly adopt the analysis in [Dieuleveut et al., 2023]. However, the condition A1 is insufficient to guarantee that $V(\boldsymbol{\theta})$ is smooth, and the mean field of (2) may not be aligned with $\nabla V(\boldsymbol{\theta})$. Instead, from A1, 2, we proceed with a descent-like lemma for the iterates of SGD-GD:

**Lemma 1.** *Under A1, 2. Suppose that the step size satisfies* $\sup_{t \geq 1} \gamma_t \leq 1/(L(1 + \sigma_1^2))$, *then for any* $t \geq 0$, *the sequence of iterates* $\{\boldsymbol{\theta}_t\}_{t \geq 0}$ *generated by SGD-GD (2) satisfies*

$$\frac{\gamma_{t+1}}{2} \|\nabla J(\boldsymbol{\theta}_t; \boldsymbol{\theta}_t)\|^2 \leq J(\boldsymbol{\theta}_t; \boldsymbol{\theta}_t) - \mathbb{E}_t[J(\boldsymbol{\theta}_{t+1}; \boldsymbol{\theta}_t)] + \frac{L}{2}\sigma_0^2\gamma_{t+1}^2. \tag{8}$$

*Proof.* Fix any $z \in \mathsf{Z}$, applying A1 and the recursion (2) lead to

$$\ell(\boldsymbol{\theta}_{t+1}; z) \leq \ell(\boldsymbol{\theta}_t; z) - \gamma_{t+1} \langle \nabla \ell(\boldsymbol{\theta}_t; z) \,|\, \nabla \ell(\boldsymbol{\theta}_t; Z_{t+1}) \rangle + \frac{L\gamma_{t+1}^2}{2} \|\nabla \ell(\boldsymbol{\theta}_t; Z_{t+1})\|^2, \tag{9}$$

for any $t \geq 0$. Note that $z \in \mathsf{Z}$ can be any fixed sample while $Z_{t+1}$ is the r.v. drawn from $\mathcal{D}(\boldsymbol{\theta}_t)$ in (2). Let $p_{\boldsymbol{\theta}_t}(z) \geq 0$ denotes the pdf of $\mathcal{D}(\boldsymbol{\theta}_t)$. We then multiply $p_{\boldsymbol{\theta}_t}(z)$ on both sides of the inequality and integrate w.r.t. $z \in \mathsf{Z}$, i.e., taking the operator $\int_{\mathsf{Z}}(\cdot)p_{\boldsymbol{\theta}_t}(z)\mathrm{d}z$. This yields

$$J(\boldsymbol{\theta}_{t+1}; \boldsymbol{\theta}_t) \leq J(\boldsymbol{\theta}_t; \boldsymbol{\theta}_t) - \gamma_{t+1} \langle \nabla J(\boldsymbol{\theta}_t; \boldsymbol{\theta}_t) \,|\, \nabla \ell(\boldsymbol{\theta}_t; Z_{t+1}) \rangle + \frac{L\gamma_{t+1}^2}{2} \|\nabla \ell(\boldsymbol{\theta}_t; Z_{t+1})\|^2, \tag{10}$$

since $\int \ell(\boldsymbol{\theta}; z)p_{\boldsymbol{\theta}_t}(z)\mathrm{d}z = J(\boldsymbol{\theta}; \boldsymbol{\theta}_t)$ according to definition (3). We next evaluate the conditional expectation, $\mathbb{E}_t[\cdot]$, on both sides of the above inequality

$$\mathbb{E}_t[J(\boldsymbol{\theta}_{t+1}; \boldsymbol{\theta}_t)] \leq J(\boldsymbol{\theta}_t; \boldsymbol{\theta}_t) - \gamma_{t+1}\|\nabla J(\boldsymbol{\theta}_t; \boldsymbol{\theta}_t)\|^2 + \frac{L\gamma_{t+1}^2}{2}\mathbb{E}_t[\|\nabla \ell(\boldsymbol{\theta}_t; Z_{t+1})\|^2]. \tag{11}$$

Using A2, we note that $\mathbb{E}_t[\|\nabla \ell(\boldsymbol{\theta}_t; Z_{t+1})\|^2] \leq \sigma_0^2 + (1 + \sigma_1^2)\|\nabla J(\boldsymbol{\theta}_t; \boldsymbol{\theta}_t)\|^2$. Reshuffling terms and using the step size condition yield the desired result (8); see §A for a detailed proof. $\square$

For sufficiently small $\gamma_{t+1} > 0$ and when $\boldsymbol{\theta}_t$ is not SPS, eq. (8) implies the descent relation $\mathbb{E}_t[J(\boldsymbol{\theta}_{t+1}; \boldsymbol{\theta}_t)] < J(\boldsymbol{\theta}_t; \boldsymbol{\theta}_t)$. This suggests that at the $t$th iteration, the function $J_t(\boldsymbol{\theta}) := J(\boldsymbol{\theta}; \boldsymbol{\theta}_t)$ may serve as a Lyapunov function for the SGD-GD scheme. Meanwhile, $J_t(\boldsymbol{\theta})$ is a *time varying* Lyapunov function. The said descent relation does not necessarily imply the convergence towards an SPS solution. Instead, the first term on the right hand side of (8) can be decomposed as:

$$\mathbb{E}[J_t(\boldsymbol{\theta}_t) - J_t(\boldsymbol{\theta}_{t+1})] = \mathbb{E}[J_t(\boldsymbol{\theta}_t) - J_{t+1}(\boldsymbol{\theta}_{t+1})] + \underbrace{\mathbb{E}[J_{t+1}(\boldsymbol{\theta}_{t+1}) - J_t(\boldsymbol{\theta}_{t+1})]}_{\text{residual}}. \tag{12}$$

The first part is a difference-of-sequence which is summable, while the second part is a residual term. The convergence of SGD-GD with non-convex losses hinges on bounding the latter residual. Taking a closer look, the residual is the difference of evaluating $\boldsymbol{\theta}_{t+1}$ on $J_t(\cdot)$ and $J_{t+1}(\cdot)$, i.e.,

$$\mathbb{E}[J_{t+1}(\boldsymbol{\theta}_{t+1}) - J_t(\boldsymbol{\theta}_{t+1})] = \mathbb{E}\left[\mathbb{E}_{Z \sim \mathcal{D}(\boldsymbol{\theta}_t), Z' \sim \mathcal{D}(\boldsymbol{\theta}_{t+1})}[\ell(\boldsymbol{\theta}_{t+1}; Z') - \ell(\boldsymbol{\theta}_{t+1}; Z)]\right]. \tag{13}$$

The above further depends on the differences between the distributions $\mathcal{D}(\boldsymbol{\theta}_t), \mathcal{D}(\boldsymbol{\theta}_{t+1})$, i.e., the *sensitivity* of the data distribution w.r.t. perturbation in $\boldsymbol{\theta}$. Next, we study sufficient conditions that imply the convergence of SGD-GD through bounding $\mathbb{E}[J_{t+1}(\boldsymbol{\theta}_{t+1}) - J_t(\boldsymbol{\theta}_{t+1})]$.

### 3.1 Sufficient Conditions for Convergence of SGD-GD

From (8), we anticipate the convergence of SGD-GD towards a biased SPS solution if it holds $\mathbb{E}[J_{t+1}(\boldsymbol{\theta}_{t+1}) - J_t(\boldsymbol{\theta}_{t+1})] = \mathcal{O}(\mathbb{E}[\|\boldsymbol{\theta}_t - \boldsymbol{\theta}_{t+1}\|]) = \mathcal{O}(\gamma_{t+1}\mathbb{E}[\|\nabla\ell(\boldsymbol{\theta}_t; Z_{t+1})\|])$. Now, as seen from (13), establishing such relation would require $\mathcal{D}(\boldsymbol{\theta})$ to satisfy a certain sensitivity criterion when subject to perturbation in $\boldsymbol{\theta}$. Our subsequent discussions are organized according to various distributional distance measures on sensitivity.

**Wasserstein-1 Sensitivity.** Our first set of conditions uses the Wasserstein-1 distance for measuring the sensitivity of data distribution. The measure is commonly used in the studies of performative prediction, e.g., as pioneered by [Perdomo et al., 2020, Mendler-Dünner et al., 2020]:

**W1.** *($\epsilon$ sensitivity w.r.t. Wasserstein-1 distance) There exists $\epsilon \geq 0$ such that*

$$W_1(\mathcal{D}(\boldsymbol{\theta}), \mathcal{D}(\boldsymbol{\theta}')) \leq \epsilon\|\boldsymbol{\theta} - \boldsymbol{\theta}'\|, \tag{14}$$

*for any $\boldsymbol{\theta}, \boldsymbol{\theta}' \in \mathbb{R}^d$. Notice that the Wasserstein-1 distance is defined as $W_1(\cdot, \cdot) := \inf_{P \in \mathcal{P}(\cdot, \cdot)} \mathbb{E}_{(z,z') \sim P}[\|z - z'\|_1]$, where $\mathcal{P}(\mathcal{D}(\boldsymbol{\theta}), \mathcal{D}(\boldsymbol{\theta}'))$ is the set of all joint distributions on $\mathsf{Z} \times \mathsf{Z}$ whose marginal distributions are $\mathcal{D}(\boldsymbol{\theta}), \mathcal{D}(\boldsymbol{\theta}')$.*

In this case, we require the loss function to be Lipschitz continuous w.r.t. shifts in the data sample $z$.

**W2.** *There exists a constant $L_0 > 0$ such that for all $z, z' \in \mathsf{Z}$, and $\boldsymbol{\theta} \in \mathbb{R}^d$,*

$$|\ell(\boldsymbol{\theta}; z) - \ell(\boldsymbol{\theta}; z')| \leq L_0 \|z - z'\|. \tag{15}$$

Our key observation is that the above conditions imply the desired Lipschitz continuity property on $J(\boldsymbol{\theta}; \cdot)$. In fact, we have:

**Lemma 2.** *Under W1, 2. For any $\boldsymbol{\theta}_1, \boldsymbol{\theta}_2, \boldsymbol{\theta} \in \mathbb{R}^d$, it holds*

$$|J(\boldsymbol{\theta}; \boldsymbol{\theta}_1) - J(\boldsymbol{\theta}; \boldsymbol{\theta}_2)| \leq L_0\epsilon \|\boldsymbol{\theta}_1 - \boldsymbol{\theta}_2\|. \tag{16}$$

The proof, which is a variant of [Drusvyatskiy and Xiao, 2023, Lemma 2.1], can be found in §B.

**TV distance Sensitivity.** Although W1 holds for a number of applications, such as the location family distributions (e.g., [Miller et al., 2021, Perdomo et al., 2020, Narang et al., 2023]), the assumption of Lipschitz loss function in W2 can be difficult to verify, especially if we want $L_0$ (and thus the Lipschitz continuity constant of $\ell(\boldsymbol{\theta}; \cdot)$ given by $L_0\epsilon$) to be small. As an alternative, we consider a slightly stronger sensitivity condition on $\mathcal{D}(\boldsymbol{\theta})$ via the total variation (TV) distance.

**C1.** *($\epsilon$ sensitivity w.r.t. TV distance) For any $\boldsymbol{\theta}, \boldsymbol{\theta}' \in \mathbb{R}^d$, there exists a constant $\epsilon \geq 0$ such that*

$$\delta_{TV}(\mathcal{D}(\boldsymbol{\theta}_1), \mathcal{D}(\boldsymbol{\theta}_2)) \leq \epsilon \|\boldsymbol{\theta} - \boldsymbol{\theta}'\|, \tag{17}$$

*where $\delta_{TV}(\cdot, \cdot)$ is the total variation distance defined as $\delta_{TV}(\mu, \nu) := \sup_{A \subset \mathsf{Z}} |\mu(A) - \nu(A)| = \frac{1}{2}\int |p_\mu(z) - p_\nu(z)| \, \mathrm{d}z$ such that $\mu, \nu$ are two probability measures supported on $\mathsf{Z}$ and $p_{(\cdot)}(z)$ denotes their probability distribution functions (p.d.f.s).*

Although C1 is slightly strengthened from W1, it allows us to relax the Lipschitz continuity assumption W2 on the loss. Particularly, we consider replacing W2 by:

**C2.** *There exists a constant $\ell_{max} \geq 0$ such that $\sup_{\boldsymbol{\theta} \in \mathbb{R}^d, z \in \mathsf{Z}} |\ell(\boldsymbol{\theta}; z)| \leq \ell_{max}$.*

The above condition requires $\ell(\cdot; \cdot)$ to be uniformly bounded. Compared to W2, it can be easier to verify and $\ell_{max}$ is typically small. For example, it holds with $\ell_{max} = 1$ for the case of sigmoid loss.

Similar to W1, 2, we observe that the above conditions imply $J(\boldsymbol{\theta}; \cdot)$ is Lipschitz continuous:

**Lemma 3.** *Under C1, 2. For any $\boldsymbol{\theta}_1, \boldsymbol{\theta}_2, \boldsymbol{\theta} \in \mathbb{R}^d$, it holds that*

$$|J(\boldsymbol{\theta}; \boldsymbol{\theta}_1) - J(\boldsymbol{\theta}; \boldsymbol{\theta}_2)| \leq 2\ell_{max}\epsilon \|\boldsymbol{\theta}_1 - \boldsymbol{\theta}_2\|. \tag{18}$$

See §C. The only difference with Lemma 2 is that (18) has a different Lipschitz constant.

**Remark 1.** *It is worth noting that in lieu of C1, Mofakhami et al. [2023] assumed the following sensitivity condition with respect to the Pearson $\chi^2$ divergence, i.e.,*

$$\chi^2(\mathcal{D}(\boldsymbol{\theta}), \mathcal{D}(\boldsymbol{\theta}')) := \int \frac{(p_{\boldsymbol{\theta}}(z) - p_{\boldsymbol{\theta}'}(z))^2}{p_{\boldsymbol{\theta}}(z)} \mathrm{d}z = \mathcal{O}(\|f_{\boldsymbol{\theta}}(\cdot) - f_{\boldsymbol{\theta}'}(\cdot)\|^2) \tag{19}$$

where $f_{\boldsymbol{\theta}}(\cdot)$ represents the output of a neural network parameterized by $\boldsymbol{\theta}$, $p_{\boldsymbol{\theta}}(\cdot)$ and $p_{\boldsymbol{\theta}'}(\cdot)$ are the probability density functions (p.d.f.s) of the induced distributions $\mathcal{D}(\boldsymbol{\theta})$ and $\mathcal{D}(\boldsymbol{\theta}')$, respectively.

Our TV distance sensitivity condition in C1 constitutes a weaker condition since for any bounded sample space $\mathsf{Z}$, the following holds:

$$W_1(\mathcal{D}(\boldsymbol{\theta}), \mathcal{D}(\boldsymbol{\theta}')) \leq \mathrm{diam}(\mathsf{Z}) \cdot \delta_{TV}(\mathcal{D}(\boldsymbol{\theta}), \mathcal{D}(\boldsymbol{\theta}')) \leq \frac{\mathrm{diam}(\mathsf{Z})}{2}\sqrt{\chi^2(\mathcal{D}(\boldsymbol{\theta}), \mathcal{D}(\boldsymbol{\theta}'))},$$

as shown in [Gibbs and Su, 2002, Sec. 2], where $\mathrm{diam}(\mathsf{Z}) := \sup_{z,z' \in \mathsf{Z}} \|z - z'\|$ denotes the diameter of the sample space.

## 3.2 Convergence of SGD-GD with Non-convex Loss

Equipped with Lemmas 2, 3, we are ready to present the convergence result for SGD-GD with smooth but non-convex losses. Observe the following theorem whose proof can be found in §D:

---

**Theorem 1.** *Under A1, 2. Let the step sizes satisfy* $\sup_{t \geq 1} \gamma_t \leq 1/(L(1 + \sigma_1^2))$. *Moreover, let*

$$\tilde{L} = L_0 \text{ if } W1, 2 \text{ hold, or } \tilde{L} = 2\ell_{max} \text{ if } C1, 2 \text{ hold.} \tag{20}$$

*Then, for any $T \geq 1$, the iterates $\{\boldsymbol{\theta}_t\}_{t \geq 0}$ generated by SGD-GD satisfy*

$$\sum_{t=0}^{T-1} \frac{\gamma_{t+1}}{4} \mathbb{E}[\|\nabla J(\boldsymbol{\theta}_t; \boldsymbol{\theta}_t)\|^2] \leq \Delta_0 + \tilde{L}\epsilon\left(\sigma_0 + (1 + \sigma_1^2)\tilde{L}\epsilon\right)\sum_{t=0}^{T-1} \gamma_{t+1} + \frac{L}{2}\sigma_0^2 \sum_{t=0}^{T-1} \gamma_{t+1}^2, \tag{21}$$

*where $\Delta_0 := J(\boldsymbol{\theta}_0; \boldsymbol{\theta}_0) - \ell_\star$ is an upper bound to the initial optimality gap of performative risk.*

---

Using a fixed step size schedule, we simplify the bound as:

**Corollary 1.** *Under A1, 2, the alternative conditions W1, 2, or C1, 2. Let $T \geq 1$ be the maximum number of iterations and set $\gamma_t = 1/\sqrt{T}$. Let $\mathsf{T}$ be a random variable chosen uniformly and independently from $\{0, 1, \cdots, T-1\}$. For any $T \geq L^2(1 + \sigma_1^2)^2$, the iterates by SGD-GD satisfy*

$$\mathbb{E}\left[\|\nabla J(\boldsymbol{\theta}_\mathsf{T}; \boldsymbol{\theta}_\mathsf{T})\|^2\right] \leq 4\left(\Delta_0 + \frac{L}{2}\sigma_0^2\right) \cdot \frac{1}{\sqrt{T}} + \underbrace{4\tilde{L}\epsilon\left(\sigma_0 + (1 + \sigma_1^2)\tilde{L}\epsilon\right)}_{\text{bias}}. \tag{22}$$

As $T \to \infty$, the first term in (22) vanishes as $\mathcal{O}(1/\sqrt{T})$ and the above shows that the SGD-GD scheme finds an $\mathcal{O}(\sigma_0\,\epsilon + (1 + \sigma_1^2)\,\epsilon^2)$-SPS solution. This yields the first convergence guarantee for performative prediction with non-convex loss via a stochastic optimization scheme.

Lastly, an interesting observation is that the bias level is controlled at $\mathcal{O}(\sigma_0\,\epsilon + (1 + \sigma_1^2)\,\epsilon^2)$. The latter estimate highlights the role of the stochastic gradient's variance. To see this, let us concentrate on the case when $\epsilon$ is small. When stochastic gradient is used such that $\sigma_0 > 0$, SGD-GD finds an $\mathcal{O}(\epsilon)$-SPS solution; while with deterministic gradient, i.e., when $\nabla \ell(\boldsymbol{\theta}_t; Z_{t+1}) = \nabla J(\boldsymbol{\theta}_t; \boldsymbol{\theta}_t)$ with $\sigma_0 = \sigma_1 = 0$, SGD-GD finds an $\mathcal{O}(\epsilon^2)$-SPS solution. Such a distinction in the bias levels indicate that a *unique property* of non-convex performative prediction where the asymptotic performance of SGD-GD is sensitive to the stochastic gradient's noise variance. Furthermore, our result suggests that adjusting the minibatch size in SGD-GD may have a significant effect on reducing the bias level since $\sigma_0, \sigma_1$ can be controlled by the latter.

**Remark 2.** *Prior analysis in [Mendler-Dünner et al., 2020, Drusvyatskiy and Xiao, 2023] showed that with $\mu$ strongly convex loss $\ell(\cdot; z)$, both the existence/uniqueness of the PS solution and the convergence of SGD-GD to the PS solution critically depend on the condition $\epsilon < \mu/L$ (in addition to our A1, 2, W1). When $\epsilon > \mu/L$, it is shown that the SGD-GD scheme may even diverge. In contrary, Theorem 1 does not exhibit such an explicit condition on $\epsilon$ for the convergence results (21), (22) to hold. This happens because our result requires the loss function itself to be Lipschitz [cf. W2] or bounded [cf. C2], which may not be satisfied by their strongly convex losses.*

## 4 Extension: Lazy Deployment Scheme with SGD

Implementing the SGD-GD scheme (2) requires deploying the latest model every time when drawing samples from $\mathcal{D}(\cdot)$. This may be difficult to realize since deploying a new classifier in real time can

be time consuming. As inspired by [Mendler-Dünner et al., 2020], this section studies an extension of (2) to the *lazy deployment* scheme where the new (prediction) models are deployed only once per several SGD updates.

To describe the extended scheme, let $K \geq 1$ denotes the epoch length of lazy deployment, we have

$$\boldsymbol{\theta}_{t,k+1} = \boldsymbol{\theta}_{t,k} - \gamma \nabla \ell(\boldsymbol{\theta}_{t,k}; Z_{t,k+1}), \text{ where } Z_{t,k+1} \sim \mathcal{D}(\boldsymbol{\theta}_t), \ k = 0, ..., K-1,$$
$$\boldsymbol{\theta}_{t+1} = \boldsymbol{\theta}_{t+1,0} = \boldsymbol{\theta}_{t,K}. \tag{23}$$

For simplicity, we focus on the case with a constant step size $\gamma > 0$. Observe that the lazy deployment scheme is a double-loop algorithm where the index $t$ denotes the number of deployments and the index $k$ denotes the SGD update. To analyze the convergence of (23), we further require the stochastic gradient to be uniformly bounded:

**A3.** *There exists a constant $G \geq 0$ such that $\sup_{\boldsymbol{\theta} \in \mathbb{R}^d, z \in \mathsf{Z}} \|\nabla \ell(\boldsymbol{\theta}; z)\| \leq G$.*

Despite being a stronger assumption, the above remains valid for practical non-convex losses, e.g., sigmoid loss. We observe the following convergence results whose proof is in §E:

---

**Theorem 2.** *Under A1, 2, 3, and the alternative conditions W1, 2, or C1, 2. Let $T \geq 1$ be the maximum number of deployments to be run, we set $\gamma = 1/(K\sqrt{T})$ and let $\mathsf{T}$ be a random variable chosen uniformly and independently from $\{0, 1, \cdots, T-1\}$. For any $T \geq L^2(1 + \sigma_1^2)^2/K^2$, the iterates generated by the lazy deployment scheme with SGD (23) satisfy:*

$$\mathbb{E}\left[\|\nabla J(\boldsymbol{\theta}_{\mathsf{T}}; \boldsymbol{\theta}_{\mathsf{T}})\|^2\right] \leq \frac{8\Delta_0}{\sqrt{T}} + \frac{4L\sigma_0^2}{K\sqrt{T}} + \frac{2LG^2}{3T} + \frac{8\tilde{L}\epsilon}{K}\left(\sqrt{2K}\sigma_0 + 2(K + \sigma_1^2)\tilde{L}\epsilon\right), \quad (24)$$

*where we recall that $\tilde{L}$ was defined in (20) and $\Delta_0 = J(\boldsymbol{\theta}_0; \boldsymbol{\theta}_0) - \ell_\star$.*

---

In (24), the first three terms decay as $\mathcal{O}(1/\sqrt{T})$ similar to SGD-GD, the last term simplifies to $\mathcal{O}((\tilde{L}\epsilon)^2 \frac{K + \sigma_1^2}{K})$. The lazy deployment scheme (23) finds an $\mathcal{O}((\tilde{L}\epsilon)^2)$-*SPS solution* when $T \to \infty, K \to \infty$, contrasting with SGD-GD which admits a bias level of $\mathcal{O}(\tilde{L}\epsilon)$.

We remark that the above result can be anticipated. During the $t$th deployment, (23) runs an SGD recursion for $\min_{\boldsymbol{\theta}} J(\boldsymbol{\theta}; \boldsymbol{\theta}_t)$ where it will find a stationary solution for the non-convex optimization as $K \to \infty$. The lazy deployment scheme resembles RRM and we expect that it may find a reduced bias SPS solution as inspired by [Mofakhami et al., 2023] which studied a similar algorithm.

## 5 Numerical Experiments

We consider two examples of performative prediction with non-convex loss based on synthetic data and real data. All simulations are performed with Pytorch on a server using a Intel Xeon 6318 CPU. Additional results can be found in §F.

**Synthetic Data with Linear Model.** We first consider a binary classification problem using linear model. To enhance robustness to outliers, we adopt the sigmoid loss function [Ertekin et al., 2010]:

$$\ell(\boldsymbol{\theta}; z) := (1 + \exp(c \cdot y\langle x \,|\, \boldsymbol{\theta}\rangle))^{-1} + (\beta/2)\|\boldsymbol{\theta}\|^2. \tag{25}$$

For small regularization $\beta > 0$, $\ell(\cdot; z)$ is smooth but non-convex. To define the data distribution, we have a set of $m$ unshifted samples $\mathcal{D}^o \equiv \{(x_i, y_i)\}_{i=1}^m$ with feature $x_i \in \mathbb{R}^d$ and label $y_i \in \{\pm 1\}$. For any $\boldsymbol{\theta} \in \mathbb{R}^d$, $\mathcal{D}(\boldsymbol{\theta})$ is a uniform distribution on $m$ *shifted samples* $\{(x_i - \epsilon_L\boldsymbol{\theta}, y_i)\}_{i=1}^m$, where $\epsilon_L > 0$ controls the shift magnitude. Applying SGD-GD to the setup yields a scheme such that A1, 2, W1 (with $\epsilon = \epsilon_L$) are satisfied, and W2 holds as $\|\boldsymbol{\theta}^t\|$ is bounded in practice. To generate the training data, the unshifted samples $\mathcal{D}^o$ are generated first as $x_i \sim \mathcal{U}[-1, 1]^d$, i.e., the uniform distribution, $\bar{y}_i = \text{sgn}(\langle x_i \,|\, \boldsymbol{\theta}^o\rangle) \in \{\pm 1\}$ such that $\boldsymbol{\theta}^o \sim \mathcal{N}(0, \boldsymbol{I})$, then a randomly selected 10% of the labels are flipped to generate the final $y_i$. Furthermore, we set $m = 800, d = 10, c = 0.1, \beta = 10^{-3}, \epsilon \in \{0, 0.1, 0.5, 2\}$. For (2), the batch size is $b = 1$ and the stepsize is $\gamma_t = \gamma = 1/\sqrt{T}$ with $T = 10^6$.

First, we validate the convergence behavior of SGD-GD in Theorem 1. In Fig. 1 (left), we compare $\|\nabla J(\boldsymbol{\theta}_t; \boldsymbol{\theta}_t)\|^2$ against the number of iteration $t$ for the SGD-GD scheme over 10 repeated runs. The shaded region indicate the 95% confidence interval. We observe that after a rapid transient stage,

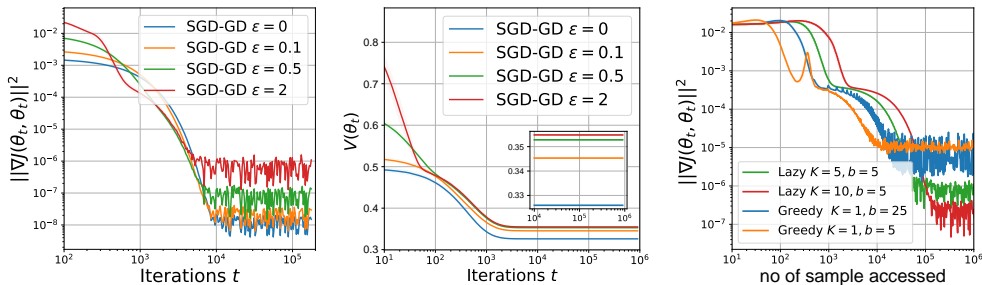

Figure 1: **Synthetic Data** (*left*) SPS measure $\|\nabla J(\boldsymbol{\theta}_t; \boldsymbol{\theta}_t)\|^2$ of SGD-GD against iteration no. $t$. (*middle*) Loss value $J(\boldsymbol{\theta}_t; \boldsymbol{\theta}_t)$ of SGD-GD against iteration no. $t$. (*right*) SPS measure $\|\nabla J(\boldsymbol{\theta}_t; \boldsymbol{\theta}_t)\|^2$ of greedy (SGD-GD) and lazy deployment against number of sample accessed. We fix $\epsilon_L = 2$.

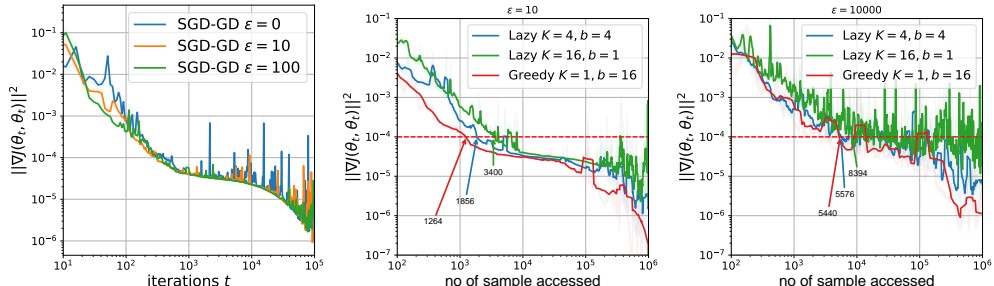

Figure 2: **Real Data with Neural Network** Benchmarking with SPS measure $\|\nabla J(\boldsymbol{\theta}_t; \boldsymbol{\theta}_t)\|^2$. (*left*) Against $t$ for SGD-GD with parameters $\epsilon_{\mathsf{NN}} \in \{0, 10, 100\}$. (*middle & right*) Against no. of samples with greedy (SGD-GD) and lazy deployment when $\epsilon_{\mathsf{NN}} = 10$ & $\epsilon_{\mathsf{NN}} = 10^4$, respectively.

the SPS stationarity $\|\nabla J(\boldsymbol{\theta}_t; \boldsymbol{\theta}_t)\|^2$ saturates and stay around a constant level, indicating that the SGD-GD converges to a biased-SPS solution. Increasing $\epsilon_L$ leads to an increased bias, corroborating with Theorem 1 that the bias level is $\mathcal{O}(\epsilon)$. Fig. 1 (middle) further evaluate the performance of the trained classifier $\boldsymbol{\theta}_t$ in terms of the performative risk value. Second, we compare the lazy deployment scheme in §4 with $K \in \{5, 10\}$ and stepsize $\gamma = 1/(K\sqrt{T})$. For fairness, we test SGD-GD with batch size of $b \in \{5, 25\}$ and compare the SPS stationarity against the number of samples accessed. The results in Fig. 1 (right) verifies Theorem 2 where increasing $K$ effectively reduces the bias level.

**Real Data with Neural Network.** Our second example deals with the task of training a neural network (NN) on the spambase Hopkins et al. [1999] dataset with $m = 4601$ samples, each with $d = 48$ features. We split the training/test sets as $8 : 2$. Our aim is to study the behavior of SGD-GD when training NN classifier. To specify (1), we let $z \equiv (x, y)$ where $x \in \mathbb{R}^d$ is the feature vector, $y \in \{0, 1\}$ is label (0 for not spam, 1 for spam). Consider the regularized binary cross entropy loss:

$$\ell(\boldsymbol{\theta}; z) \equiv \tilde{\ell}(f_{\boldsymbol{\theta}}(x); y) = -y \log(f_{\boldsymbol{\theta}}(x)) - (1 - y) \log(1 - f_{\boldsymbol{\theta}}(x)) + (\beta/2) \|\boldsymbol{\theta}\|^2, \quad (26)$$

where $f_{\boldsymbol{\theta}}(x)$ denotes the NN classifier. The unshifted data is denoted by $\mathcal{D}^o = \{(x_i, y_i)\}_{i=1}^m$. Sampling from the shifted data distribution $\mathcal{D}(\boldsymbol{\theta})$ is achieved through (i) uniformly draw a sample $\bar{z} \equiv (\bar{x}, \bar{y})$ from $\mathcal{D}^o$, (ii) maximize the following utility function:

$$x = \arg\max_{x'} U(x'; \bar{x}, \boldsymbol{\theta}) := -f_{\boldsymbol{\theta}}(x') - \frac{1}{2\epsilon_{\mathsf{NN}}} \|x' - \bar{x}\|^2, \quad (27)$$

to get $z \equiv (x, \bar{y}) \sim \mathcal{D}(\boldsymbol{\theta})$. In practice, we take the approximation $x \approx \bar{x} - \epsilon_{\mathsf{NN}} \nabla_x f_{\boldsymbol{\theta}}(\bar{x})$.

In our experiment, we set $\epsilon_{\mathsf{NN}} \in \{0, 10, 100\}$, batch size as $b = 8$. For SGD-GD, we use $\gamma_t = \gamma = 200/\sqrt{T}$ and for lazy deployment, we use $\gamma = 200/(K\sqrt{T})$ with $T = 10^5$. The NN encoded in $f_{\boldsymbol{\theta}}(x)$ consists of three fully-connected layers with $\tanh$ activation and a sigmoid output layer, i.e.,

$$f_{\boldsymbol{\theta}}(x) = Sigmoid(\boldsymbol{\theta}_{(1)}^\top \cdot \tanh(\boldsymbol{\theta}_{(2)}^\top \cdot \tanh(\boldsymbol{\theta}_{(3)}^\top x))),$$

where $\boldsymbol{\theta}_{(i)} := [w_{(i)}; b_{(i)}]$ concatenates the weight and bias for each layer with $d_1 = 10, d_2 = 50, d_3 = 57$ neurons, making a total of $d = 3421$ parameters for $\boldsymbol{\theta}$. For the training, we initialize these parameters as $\mathcal{N}(0, 1)$ for weights and constant values for the biases.

Fig. 2 (left) compares the SPS measure $\|\nabla J(\boldsymbol{\theta}_t; \boldsymbol{\theta}_t)\|^2$ against the iteration number $t$ using SGD-GD. As observed, SGD-GD converges to a near SPS solution and the behavior seems to be insensitive to $\epsilon_{\mathsf{NN}}$. We speculate that this is due to the shift model (27) but would relegate its study to future work. Fig. 2 (middle & right) compare the greedy and lazy deployment schemes with $\epsilon_{\mathsf{NN}} \in \{10, 10^4\}$ against the number of samples used. Compared to SGD-GD, lazy deployment performs relatively better as $\epsilon_{\mathsf{NN}} \uparrow$, as seen from the no. of samples needed to reach $\|\nabla J(\boldsymbol{\theta}_t; \boldsymbol{\theta}_t)\|^2 = 10^{-4}$ in the plots. This agrees with (24) which shows the dominant term as $\mathcal{O}(\epsilon^2)$ and $\epsilon$ is related to $\epsilon_{\mathsf{NN}}$.

## 6   Conclusions

This paper provides the first study on the performative prediction problem with smooth but possibly non-convex loss. We proposed a stationary performative stable (SPS) condition which is the counterpart of performative stable condition used with strongly convex loss. Using the SPS solution concept, we studied the convergence of greedy deployment and lazy deployment schemes with SGD. We prove that SGD-GD finds a biased, $\mathcal{O}(\epsilon)$-SPS solution, while the lazy deployment scheme finds a reduced bias SPS solution when the lazy deployment epoch is large. As an initial work on this subclass of problems, our findings can lead to more general analysis on algorithms under the non-convex performative prediction framework.

## Acknowledgement

The project was supported in part by CUHK Direct Grant #4055208. The authors would like to thank the anonymous reviewer for pointing out the possibility of extending our convergence analysis to sensitivity measures defined by the TV distance.

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

# A    Proof of Lemma 1

*Proof.* Under A1, for any fixed $z \in \mathsf{Z}$, we have that

$$\ell(\boldsymbol{\theta}_{t+1}; z) \leq \ell(\boldsymbol{\theta}_t; z) + \langle \nabla\ell(\boldsymbol{\theta}_t; z) \,|\, \boldsymbol{\theta}_{t+1} - \boldsymbol{\theta}_t \rangle + \frac{L}{2} \|\boldsymbol{\theta}_{t+1} - \boldsymbol{\theta}_t\|^2$$

$$\leq \ell(\boldsymbol{\theta}_t; z) - \gamma_{t+1} \langle \nabla\ell(\boldsymbol{\theta}_t; z) \,|\, \nabla\ell(\boldsymbol{\theta}_t; Z_{t+1}) \rangle + \frac{L\gamma_{t+1}^2}{2} \|\nabla\ell(\boldsymbol{\theta}_t; Z_{t+1})\|^2,$$

where the second inequality is due to the update rule of (2) as we recall that $Z_{t+1} \sim \mathcal{D}(\boldsymbol{\theta}_t)$. Taking integration on $z$ with weights given by the p.d.f. of $\mathcal{D}(\boldsymbol{\theta}_t)$, i.e., $\int (\cdot) p_{\boldsymbol{\theta}_t}(z) \mathrm{d}z$, on both sides of above inequality leads to

$$J(\boldsymbol{\theta}_{t+1}; \boldsymbol{\theta}_t) \leq J(\boldsymbol{\theta}_t; \boldsymbol{\theta}_t) - \gamma_{t+1} \langle \nabla J(\boldsymbol{\theta}_t; \boldsymbol{\theta}_t) \,|\, \nabla\ell(\boldsymbol{\theta}_t; Z_{t+1}) \rangle + \frac{L}{2}\gamma_{t+1}^2 \|\nabla\ell(\boldsymbol{\theta}_t; Z_{t+1})\|^2.$$

As $\mathbb{E}_t[\nabla\ell(\boldsymbol{\theta}_t; Z_{t+1})] = \nabla J(\boldsymbol{\theta}_t; \boldsymbol{\theta}_t)$, taking the conditional expectation $\mathbb{E}_t[\cdot]$ on both sides yield

$$\mathbb{E}_t\left[J(\boldsymbol{\theta}_{t+1}; \boldsymbol{\theta}_t)\right] \leq J(\boldsymbol{\theta}_t; \boldsymbol{\theta}_t) - \gamma_{t+1} \|\nabla J(\boldsymbol{\theta}_t; \boldsymbol{\theta}_t)\|^2 + \frac{L}{2}\gamma_{t+1}^2 \mathbb{E}_t\left[\|\nabla\ell(\boldsymbol{\theta}_t; Z_{t+1})\|^2\right] \tag{28}$$

$$\overset{(a)}{=} J(\boldsymbol{\theta}_t; \boldsymbol{\theta}_t) - \gamma_{t+1} \|\nabla J(\boldsymbol{\theta}_t; \boldsymbol{\theta}_t)\|^2$$

$$+ \frac{L}{2}\gamma_{t+1}^2 \left(\mathbb{E}_t \|\nabla\ell(\boldsymbol{\theta}_t; Z_{t+1}) - \nabla J(\boldsymbol{\theta}_t; \boldsymbol{\theta}_t)\|^2 + \|\nabla J(\boldsymbol{\theta}_t; \boldsymbol{\theta}_t)\|^2\right),$$

$$\overset{(b)}{\leq} J(\boldsymbol{\theta}_t; \boldsymbol{\theta}_t) - \gamma_{t+1} \|\nabla J(\boldsymbol{\theta}_t; \boldsymbol{\theta}_t)\|^2 + \frac{L}{2}\gamma_{t+1}^2 \left(\sigma_0^2 + (1 + \sigma_1^2) \|\nabla J(\boldsymbol{\theta}; \boldsymbol{\theta}_t)\|^2\right),$$

where $(a)$ used A2 and the property:

$$\mathbb{E}_t\left[\|\nabla\ell(\boldsymbol{\theta}_t; Z_{t+1})\|^2\right] = \mathbb{E}_t\left[\|\nabla\ell(\boldsymbol{\theta}_t; Z_{t+1}) - \nabla J(\boldsymbol{\theta}_t; \boldsymbol{\theta}_t)\|^2\right] + \|\nabla J(\boldsymbol{\theta}_t; \boldsymbol{\theta}_t)\|^2,$$

and $(b)$ is due to the variance bound in A2. Rearranging terms in (28) leads to

$$\left(1 - \frac{L}{2}(1 + \sigma_1^2)\gamma_{t+1}\right)\gamma_{t+1} \|\nabla J(\boldsymbol{\theta}_t; \boldsymbol{\theta}_t)\|^2 \leq J(\boldsymbol{\theta}_t; \boldsymbol{\theta}_t) - \mathbb{E}_t[J(\boldsymbol{\theta}_{t+1}; \boldsymbol{\theta}_t)] + \frac{L}{2}\sigma_0^2\gamma_{t+1}^2.$$

The step size condition implies $1 - \frac{L}{2}(1 + \sigma_1^2)\gamma_{t+1} \geq 1/2$. This concludes the proof. $\qquad\square$

# B    Proof of Lemma 2

*Proof.* Our proof is modified from Lemma 2.1 of [Drusvyatskiy and Xiao, 2023]. By W2, since $\ell(\boldsymbol{\theta}; z)$ is $L_0$-Lipchitz in $z$, we have

$$|J(\boldsymbol{\theta}; \boldsymbol{\theta}_1) - J(\boldsymbol{\theta}; \boldsymbol{\theta}_2)| = |\mathbb{E}_{Z \sim \mathcal{D}(\boldsymbol{\theta}_1)} \ell(\boldsymbol{\theta}; Z) - \mathbb{E}_{Z' \sim \mathcal{D}(\boldsymbol{\theta}_2)} \ell(\boldsymbol{\theta}; Z')| \leq L_0 W_1(\mathcal{D}(\boldsymbol{\theta}_1), \mathcal{D}(\boldsymbol{\theta}_2)).$$

Applying W1 gives

$$|J(\boldsymbol{\theta}; \boldsymbol{\theta}_1) - J(\boldsymbol{\theta}; \boldsymbol{\theta}_2)| \leq L_0\epsilon \|\boldsymbol{\theta}_1 - \boldsymbol{\theta}_2\|,$$

which finishes the proof. $\qquad\square$

# C    Proof of Lemma 3

*Proof.* Under C1 & C2, we observe

$$|J(\boldsymbol{\theta}, \boldsymbol{\theta}_1) - J(\boldsymbol{\theta}, \boldsymbol{\theta}_2)| = \left|\int \ell(\boldsymbol{\theta}; z)(p_{\boldsymbol{\theta}_1}(z) - p_{\boldsymbol{\theta}_2}(z))\mathrm{d}z\right|$$

$$\overset{(a)}{\leq} \int |\ell(\boldsymbol{\theta}; z)| \cdot |p_{\boldsymbol{\theta}_1}(z) - p_{\boldsymbol{\theta}_2}(z)| \, \mathrm{d}z$$

$$\leq \ell_{max} \cdot \int |p_{\boldsymbol{\theta}_1}(z) - p_{\boldsymbol{\theta}_2}(z)| \, \mathsf{d}(z)$$

$$\leq \ell_{max} \cdot 2\delta_{TV}(\mathcal{D}(\boldsymbol{\theta}_1), \mathcal{D}(\boldsymbol{\theta}_2))$$

$$\overset{(b)}{\leq} 2\ell_{max}\epsilon \|\boldsymbol{\theta}_1 - \boldsymbol{\theta}_2\|.$$

where (a) is due to the Cauchy-Schwarz inequality, (b) is due to the stated assumptions C1. $\qquad\square$

# D  Proof of Theorem 1

*Proof.* We recall from Lemma 1 the following relation:

$$\frac{\gamma_{t+1}}{2}\|\nabla J(\boldsymbol{\theta}_t;\boldsymbol{\theta}_t)\|^2 \leq \mathbb{E}_t[J(\boldsymbol{\theta}_t;\boldsymbol{\theta}_t) - J(\boldsymbol{\theta}_{t+1};\boldsymbol{\theta}_t)] + \frac{L}{2}\sigma_0^2\gamma_{t+1}^2. \tag{29}$$

We notice that Lemmas 2, 3 imply

$$|J(\bar{\boldsymbol{\theta}};\boldsymbol{\theta}) - J(\bar{\boldsymbol{\theta}};\boldsymbol{\theta}')| \leq \tilde{L}\epsilon\,\|\boldsymbol{\theta}-\boldsymbol{\theta}'\|,$$

where $\tilde{L} = L_0$ if W1, 2 hold, or $\tilde{L} = \ell_{max}$ if C1, 2 hold. Subsequently, the first term on the right hand side of (29) can be bounded by

$$\mathbb{E}_t[J(\boldsymbol{\theta}_t;\boldsymbol{\theta}_t) - J(\boldsymbol{\theta}_{t+1};\boldsymbol{\theta}_t)] \leq \mathbb{E}_t[J(\boldsymbol{\theta}_t;\boldsymbol{\theta}_t) - J(\boldsymbol{\theta}_{t+1};\boldsymbol{\theta}_{t+1})] + \mathbb{E}_t[|J(\boldsymbol{\theta}_{t+1};\boldsymbol{\theta}_{t+1}) - J(\boldsymbol{\theta}_{t+1};\boldsymbol{\theta}_t)|]$$

$$\leq \mathbb{E}_t[J(\boldsymbol{\theta}_t;\boldsymbol{\theta}_t) - J(\boldsymbol{\theta}_{t+1};\boldsymbol{\theta}_{t+1})] + \tilde{L}\epsilon\,\mathbb{E}_t[\|\boldsymbol{\theta}_{t+1} - \boldsymbol{\theta}_t\|]$$

$$= \mathbb{E}_t[J(\boldsymbol{\theta}_t;\boldsymbol{\theta}_t) - J(\boldsymbol{\theta}_{t+1};\boldsymbol{\theta}_{t+1})] + \gamma_{t+1}\tilde{L}\epsilon\,\mathbb{E}_t[\|\nabla\ell(\boldsymbol{\theta}_t;Z_{t+1})\|].$$

Notice that

$$\gamma_{t+1}\tilde{L}\epsilon\,\mathbb{E}_t\left[\|\nabla\ell(\boldsymbol{\theta}_t;Z_{t+1})\|\right] \overset{(a)}{\leq} \gamma_{t+1}\tilde{L}\epsilon\sqrt{\mathbb{E}_t\left[\|\nabla\ell(\boldsymbol{\theta}_t;Z_{t+1})\|^2\right]}$$

$$\overset{(b)}{\leq} \gamma_{t+1}\tilde{L}\epsilon\left(\sigma_0 + \sqrt{1+\sigma_1^2}\,\|\nabla J(\boldsymbol{\theta}_t;\boldsymbol{\theta}_t)\|\right)$$

$$\overset{(c)}{\leq} \gamma_{t+1}\tilde{L}\epsilon\left(\sigma_0 + (1+\sigma_1^2)\tilde{L}\epsilon + \frac{1}{4\tilde{L}\epsilon}\|\nabla J(\boldsymbol{\theta}_t;\boldsymbol{\theta}_t)\|^2\right),$$

where $(a)$ is due to the Cauchy-Schwarz inequality $\mathbb{E}[\|X\|] \leq \sqrt{\mathbb{E}[\|X\|^2]}$, $(b)$ is due to the chain:

$$\mathbb{E}_t\left[\|\nabla\ell(\boldsymbol{\theta}_t;Z_{t+1})\|^2\right] = \|\nabla J(\boldsymbol{\theta}_t;\boldsymbol{\theta}_t)\|^2 + \mathbb{E}_t\left[\|\nabla\ell(\boldsymbol{\theta}_t;Z_{t+1}) - \nabla J(\boldsymbol{\theta}_t;\boldsymbol{\theta}_t)\|^2\right]$$

$$\leq \sigma_0^2 + (1+\sigma_1^2)\|\nabla J(\boldsymbol{\theta}_t;\boldsymbol{\theta}_t)\|^2 \leq \left(\sigma_0 + \sqrt{1+\sigma_1^2}\|\nabla J(\boldsymbol{\theta}_t;\boldsymbol{\theta}_t)\|\right)^2$$

and $(c)$ is due to the Young's inequality. Substituting back into (29) gives

$$\frac{\gamma_{t+1}}{4}\|\nabla J(\boldsymbol{\theta}_t;\boldsymbol{\theta}_t)\|^2 \leq \mathbb{E}_t[J(\boldsymbol{\theta}_t;\boldsymbol{\theta}_t) - J(\boldsymbol{\theta}_{t+1};\boldsymbol{\theta}_{t+1})] + \gamma_{t+1}\tilde{L}\epsilon\left(\sigma_0 + (1+\sigma_1^2)\tilde{L}\epsilon\right) + \frac{L}{2}\sigma_0^2\gamma_{t+1}^2.$$

Notice that taking full expectation and summing both sides of the inequality from $t = 0$ to $T-1$ yields the theorem. $\qquad\square$

# E  Proof of Theorem 2

*Proof.* The first steps of our proof resemble that of Lemma 1 and is repeated here for completeness. Under A1, for any fixed $z \in \mathsf{Z}$, we have that

$$\ell(\boldsymbol{\theta}_{t,k+1};z) \leq \ell(\boldsymbol{\theta}_{t,k};z) + \langle\nabla\ell(\boldsymbol{\theta}_{t,k};z)\,|\,\boldsymbol{\theta}_{t,k+1} - \boldsymbol{\theta}_{t,k}\rangle + \frac{L}{2}\|\boldsymbol{\theta}_{t,k+1} - \boldsymbol{\theta}_{t,k}\|^2$$

$$\leq \ell(\boldsymbol{\theta}_{t,k};z) - \gamma\langle\nabla\ell(\boldsymbol{\theta}_{t,k};z)\,|\,\nabla\ell(\boldsymbol{\theta}_{t,k};Z_{t,k+1})\rangle + \frac{L\gamma^2}{2}\|\nabla\ell(\boldsymbol{\theta}_{t,k};Z_{t,k+1})\|^2,$$

where the second inequality is due to the update rule of (2) as we recall that $Z_{t+1} \sim \mathcal{D}(\boldsymbol{\theta}_t)$. Taking integration on $z$ with weights given by the p.d.f. of $\mathcal{D}(\boldsymbol{\theta}_t)$, i.e., $\int(\cdot)p_{\boldsymbol{\theta}_t}(z)\mathrm{d}z$, on both sides of above inequality leads to

$$J(\boldsymbol{\theta}_{t,k+1};\boldsymbol{\theta}_{t,0}) \leq J(\boldsymbol{\theta}_{t,k};\boldsymbol{\theta}_{t,0}) - \gamma\langle\nabla J(\boldsymbol{\theta}_{t,k};\boldsymbol{\theta}_{t,0})\,|\,\nabla\ell(\boldsymbol{\theta}_{t,k};Z_{t,k+1})\rangle + \frac{L}{2}\gamma^2\|\nabla\ell(\boldsymbol{\theta}_{t,k};Z_{t,k+1})\|^2.$$

As $Z_{t,k+1} \sim \mathcal{D}(\boldsymbol{\theta}_{t,0})$, we have $\mathbb{E}_{t,k}[\nabla\ell(\boldsymbol{\theta}_{t,k};Z_{t,k+1})] = \nabla J(\boldsymbol{\theta}_{t,k};\boldsymbol{\theta}_{t,0})$, where $\mathbb{E}_{t,k}[\cdot]$ denotes the conditional expectation on the filtration

$$\mathcal{F}_{t,k} = \sigma(\{\boldsymbol{\theta}_0,\boldsymbol{\theta}_{0,1},\cdots,\boldsymbol{\theta}_{0,K},\boldsymbol{\theta}_{1,1},\cdots,\boldsymbol{\theta}_t,\boldsymbol{\theta}_{t,1},\cdots,\boldsymbol{\theta}_{t,k}\}).$$

Taking the conditional expectation $\mathbb{E}_{t,k}[\cdot]$ on both sides yield

$$\mathbb{E}_{t,k}\left[J(\boldsymbol{\theta}_{t,k+1};\boldsymbol{\theta}_{t,0})\right] \leq J(\boldsymbol{\theta}_{t,k};\boldsymbol{\theta}_{t,0}) - \gamma\left\|\nabla J(\boldsymbol{\theta}_{t,k};\boldsymbol{\theta}_{t,0})\right\|^2 + \frac{L\gamma^2}{2}\mathbb{E}_{t,k}\left\|\nabla\ell(\boldsymbol{\theta}_{t,k};Z_{t,k+1})\right\|^2 \qquad (30)$$

$$\overset{(a)}{=} J(\boldsymbol{\theta}_{t,k};\boldsymbol{\theta}_{t,0}) - \gamma\left\|\nabla J(\boldsymbol{\theta}_{t,k};\boldsymbol{\theta}_{t,0})\right\|^2$$
$$+ \frac{L}{2}\gamma^2\left(\mathbb{E}_{t,k}\left\|\nabla\ell(\boldsymbol{\theta}_{t,k};Z_{t,k+1}) - \nabla J(\boldsymbol{\theta}_{t,k};\boldsymbol{\theta}_{t,0})\right\|^2 + \left\|\nabla J(\boldsymbol{\theta}_{t,k};\boldsymbol{\theta}_{t,0})\right\|^2\right),$$

$$\overset{(b)}{\leq} J(\boldsymbol{\theta}_{t,k};\boldsymbol{\theta}_{t,0}) - \gamma\left\|\nabla J(\boldsymbol{\theta}_{t,k};\boldsymbol{\theta}_{t,0})\right\|^2 + \frac{L}{2}\gamma^2\left(\sigma_0^2 + (1+\sigma_1^2)\left\|\nabla J(\boldsymbol{\theta}_{t,k};\boldsymbol{\theta}_{t,0})\right\|^2\right),$$

where $(a)$ used A2 and the property:

$$\mathbb{E}_{t,k}\left[\left\|\nabla\ell(\boldsymbol{\theta}_{t,k};Z_{t,k+1})\right\|^2\right] = \mathbb{E}_{t,k}\left[\left\|\nabla\ell(\boldsymbol{\theta}_{t,k};Z_{t,k+1}) - \nabla J(\boldsymbol{\theta}_{t,k};\boldsymbol{\theta}_{t,0})\right\|^2\right] + \left\|\nabla J(\boldsymbol{\theta}_{t,k};\boldsymbol{\theta}_{t,0})\right\|^2,$$

and $(b)$ is due to the variance bound in A2. Rearranging terms in (30),

$$\left(1 - \frac{L}{2}(1+\sigma_1^2)\gamma\right)\gamma\left\|\nabla J(\boldsymbol{\theta}_{t,k};\boldsymbol{\theta}_{t,0})\right\|^2 \leq J(\boldsymbol{\theta}_{t,k};\boldsymbol{\theta}_{t,0}) - \mathbb{E}_{t,k}[J(\boldsymbol{\theta}_{t,k+1};\boldsymbol{\theta}_{t,0})] + \frac{L}{2}\sigma_0^2\gamma^2. \quad (31)$$

The step size condition implies $1 - \frac{L}{2}(1+\sigma_1^2)\gamma \geq \frac{1}{2}$.

$$\frac{\gamma}{2}\left\|\nabla J(\boldsymbol{\theta}_{t,k};\boldsymbol{\theta}_{t,0})\right\|^2 \leq J(\boldsymbol{\theta}_{t,k};\boldsymbol{\theta}_{t,0}) - \mathbb{E}_{t,k}[J(\boldsymbol{\theta}_{t,k+1};\boldsymbol{\theta}_{t,0})] + \frac{L}{2}\sigma_0^2\gamma^2.$$

Taking summation on $k$ from 0 to $K-1$ leads to

$$\frac{\gamma}{2}\sum_{k=0}^{K-1}\left\|\nabla J(\boldsymbol{\theta}_{t,k};\boldsymbol{\theta}_{t,0})\right\|^2 \leq J(\boldsymbol{\theta}_{t,0};\boldsymbol{\theta}_{t,0}) - \mathbb{E}_{t,k}[J(\boldsymbol{\theta}_{t+1,0};\boldsymbol{\theta}_{t,0})] + \frac{LK}{2}\sigma_0^2\gamma^2. \qquad (32)$$

Recall that $\boldsymbol{\theta}_{t+1,0} = \boldsymbol{\theta}_{t+1} = \boldsymbol{\theta}_{t,K}$. Subsequently, the first term on the right hand side of (32) can be bounded by

$$\mathbb{E}_t[J(\boldsymbol{\theta}_t;\boldsymbol{\theta}_t) - J(\boldsymbol{\theta}_{t+1};\boldsymbol{\theta}_t)] \leq \mathbb{E}_t[J(\boldsymbol{\theta}_t;\boldsymbol{\theta}_t) - J(\boldsymbol{\theta}_{t+1};\boldsymbol{\theta}_{t+1})] + \mathbb{E}_t[|J(\boldsymbol{\theta}_{t+1};\boldsymbol{\theta}_{t+1}) - J(\boldsymbol{\theta}_{t+1};\boldsymbol{\theta}_t)|]$$
$$\leq \mathbb{E}_t[J(\boldsymbol{\theta}_t;\boldsymbol{\theta}_t) - J(\boldsymbol{\theta}_{t+1};\boldsymbol{\theta}_{t+1})] + \tilde{L}\epsilon\,\mathbb{E}_t[\|\boldsymbol{\theta}_{t+1} - \boldsymbol{\theta}_t\|]$$
$$= \mathbb{E}_t[J(\boldsymbol{\theta}_t;\boldsymbol{\theta}_t) - J(\boldsymbol{\theta}_{t+1};\boldsymbol{\theta}_{t+1})] + \gamma\tilde{L}\epsilon\,\mathbb{E}_t\left[\left\|\sum_{k=0}^{K-1}\nabla\ell(\boldsymbol{\theta}_{t,k};Z_{t,k+1})\right\|\right].$$

Notice that through a careful use of A2 and the independence between stochastic gradients, we have

$$\mathbb{E}_t\left[\left\|\sum_{k=0}^{K-1}\nabla\ell(\boldsymbol{\theta}_{t,k};Z_{t,k+1})\right\|\right] \leq \sqrt{\mathbb{E}_t\left[\left\|\sum_{k=0}^{K-1}\nabla\ell(\boldsymbol{\theta}_{t,k};Z_{t,k+1})\right\|^2\right]}$$

$$\leq \sqrt{2\mathbb{E}_t\left[\left\|\sum_{k=0}^{K-1}(\nabla\ell(\boldsymbol{\theta}_{t,k};Z_{t,k+1}) - \nabla J(\boldsymbol{\theta}_{t,k};\boldsymbol{\theta}_t))\right\|^2\right] + 2\mathbb{E}_t\left[\left\|\sum_{k=0}^{K-1}\nabla J(\boldsymbol{\theta}_{t,k};\boldsymbol{\theta}_t)\right\|^2\right]}$$

$$= \sqrt{2\sum_{k=0}^{K-1}\mathbb{E}_t\left[\left\|\nabla\ell(\boldsymbol{\theta}_{t,k};Z_{t,k+1}) - \nabla J(\boldsymbol{\theta}_{t,k};\boldsymbol{\theta}_t)\right\|^2\right] + 2\mathbb{E}_t\left[\left\|\sum_{k=0}^{K-1}\nabla J(\boldsymbol{\theta}_{t,k};\boldsymbol{\theta}_t)\right\|^2\right]} \qquad (33)$$

$$\leq \sqrt{2K\sigma_0^2 + 2\sigma_1^2\sum_{k=0}^{K-1}\mathbb{E}_t\left[\left\|\nabla J(\boldsymbol{\theta}_{t,k};\boldsymbol{\theta}_t)\right\|^2\right] + 2\mathbb{E}_t\left[\left\|\sum_{k=0}^{K-1}\nabla J(\boldsymbol{\theta}_{t,k};\boldsymbol{\theta}_t)\right\|^2\right]}$$

$$\leq \sqrt{2K\sigma_0^2 + 2(K+\sigma_1^2)\sum_{k=0}^{K-1}\mathbb{E}_t\left[\left\|\nabla J(\boldsymbol{\theta}_{t,k};\boldsymbol{\theta}_t)\right\|^2\right]}.$$

Using $\sqrt{a^2 + b^2} \leq a + b$ for $a, b \geq 0$, we have

$$\mathbb{E}_t \left[ \left\| \sum_{k=0}^{K-1} \nabla \ell(\boldsymbol{\theta}_{t,k}; Z_{t,k+1}) \right\| \right] \leq \sqrt{2K}\sigma_0 + \sqrt{2(K+\sigma_1^2)} \sqrt{\sum_{k=0}^{K-1} \mathbb{E}_t \left[ \|\nabla J(\boldsymbol{\theta}_{t,k}; \boldsymbol{\theta}_t)\|^2 \right]} \tag{34}$$

$$\leq \sqrt{2K}\sigma_0 + 2(K+\sigma_1^2)\tilde{L}\epsilon + \frac{1}{4\tilde{L}\epsilon} \sum_{k=0}^{K-1} \mathbb{E}_t \left[ \|\nabla J(\boldsymbol{\theta}_{t,k}; \boldsymbol{\theta}_t)\|^2 \right],$$

where the last inequality used the property $\sqrt{ax} \leq \frac{ca}{2} + \frac{x}{2c}$ for any $c > 0$. Substituting above results to (32) and taking full expectation on both sides give us

$$\frac{\gamma}{4} \sum_{k=0}^{K-1} \mathbb{E} \|\nabla J(\boldsymbol{\theta}_{t,k}, \boldsymbol{\theta}_{t,0})\|^2 \leq \mathbb{E}\left[ J(\boldsymbol{\theta}_t, \boldsymbol{\theta}_t) - J(\boldsymbol{\theta}_{t+1}, \boldsymbol{\theta}_{t+1}) \right] \tag{35}$$

$$+ \gamma\tilde{L}\epsilon \left( \sqrt{2K}\sigma_0 + 2(K+\sigma_1^2)\tilde{L}\epsilon \right) + \frac{LK}{2}\sigma_0^2\gamma^2.$$

Next, we lower bound the left hand side by observing:

$$\sum_{k=0}^{K-1} \mathbb{E} \|\nabla J(\boldsymbol{\theta}_{t,k}; \boldsymbol{\theta}_{t,0})\|^2 \overset{(a)}{\geq} \sum_{k=0}^{K-1} \mathbb{E} \left[ \frac{1}{2} \|\nabla J(\boldsymbol{\theta}_t, \boldsymbol{\theta}_t)\|^2 - \|\nabla J(\boldsymbol{\theta}_{t,k}; \boldsymbol{\theta}_t) - \nabla J(\boldsymbol{\theta}_t, \boldsymbol{\theta}_t)\|^2 \right]$$

$$\overset{(b)}{\geq} \frac{1}{2}K\mathbb{E} \|\nabla J(\boldsymbol{\theta}_t, \boldsymbol{\theta}_t)\|^2 - L \sum_{k=0}^{K-1} \mathbb{E} \|\boldsymbol{\theta}_{t,k} - \boldsymbol{\theta}_t\|^2$$

$$\overset{(c)}{=} \frac{1}{2}K\mathbb{E} \|\nabla J(\boldsymbol{\theta}_t, \boldsymbol{\theta}_t)\|^2 - L \sum_{k=0}^{K-1} \mathbb{E} \left\| \sum_{\ell=0}^{k-1} \gamma\nabla\ell(\boldsymbol{\theta}_{t,\ell}; Z_{t,\ell}) \right\|^2$$

$$\geq \frac{1}{2}K\mathbb{E} \|\nabla J(\boldsymbol{\theta}_t, \boldsymbol{\theta}_t)\|^2 - L\gamma^2 \sum_{k=0}^{K-1} k \sum_{\ell=0}^{k-1} \mathbb{E} \|\nabla\ell(\boldsymbol{\theta}_{t,\ell}; Z_{t,\ell})\|^2$$

$$\overset{(d)}{\geq} \frac{1}{2}K\mathbb{E} \|\nabla J(\boldsymbol{\theta}_t, \boldsymbol{\theta}_t)\|^2 - L\gamma^2 \sum_{k=0}^{K-1} k \sum_{\ell=0}^{k-1} G^2$$

$$= \frac{1}{2}K\mathbb{E} \|\nabla J(\boldsymbol{\theta}_t, \boldsymbol{\theta}_t)\|^2 - LG^2\gamma^2 \cdot \frac{K(K-1)(2K-1)}{6}$$

$$\geq \frac{1}{2}K\mathbb{E} \|\nabla J(\boldsymbol{\theta}_t, \boldsymbol{\theta}_t)\|^2 - LG^2\gamma^2 \cdot \frac{K^3}{3},$$

where $(a)$ is due to the fact that $\|a\|^2 \geq \frac{1}{2}\|a+b\|^2 - \|b\|^2$, for any $a, b \in \mathbb{R}^n$, $(b)$ is due to A1, $(c)$ is obtained from the updating rule (23). In $(d)$, we used the additional assumption A3. The last chain is due to $\sum_{k=0}^{K-1} k^2 = \frac{K(K-1)(2K-1)}{6} \leq \frac{K^3}{3}$, when $K \geq 1$. Substituting the above lower bound to (35) and rearrange terms lead to

$$\frac{\gamma K}{8} \mathbb{E} \|\nabla J(\boldsymbol{\theta}_t, \boldsymbol{\theta}_t)\|^2 \leq \mathbb{E}\left[ J(\boldsymbol{\theta}_t, \boldsymbol{\theta}_t) - J(\boldsymbol{\theta}_{t+1}, \boldsymbol{\theta}_{t+1}) \right] + \frac{1}{12}\gamma^3 LG^2 K^3$$

$$+ \gamma\tilde{L}\epsilon \left( \sqrt{2K}\sigma_0 + 2(K+\sigma_1^2)\tilde{L}\epsilon \right) + \frac{LK}{2}\sigma_0^2\gamma^2.$$

Taking summation from $t = 0, 1, \cdots, T-1$ gives us

$$\frac{\gamma K}{8} \sum_{t=0}^{T-1} \mathbb{E} \|\nabla J(\boldsymbol{\theta}_t, \boldsymbol{\theta}_t)\|^2 \leq \mathbb{E}\left[ J(\boldsymbol{\theta}_0, \boldsymbol{\theta}_0) - J(\boldsymbol{\theta}_T, \boldsymbol{\theta}_T) \right] + \frac{T}{12}\gamma^3 LG^2 K^3$$

$$+ \gamma T\tilde{L}\epsilon \left( \sqrt{2K}\sigma_0 + 2(K+\sigma_1^2)\tilde{L}\epsilon \right) + \frac{TLK}{2}\sigma_0^2\gamma^2.$$

Dividing $\gamma KT/8$ on both sides, we have

$$\frac{1}{T} \sum_{t=0}^{T-1} \mathbb{E} \|\nabla J(\boldsymbol{\theta}_t, \boldsymbol{\theta}_t)\|^2 \leq \frac{8\Delta_0}{\gamma KT} + 4L\sigma_0^2\gamma + \frac{2}{3}\gamma^2 LG^2 K^2 + \frac{8\tilde{L}\epsilon}{K} \left( \sqrt{2K}\sigma_0 + 2(K+\sigma_1^2)\tilde{L}\epsilon \right),$$

where we recall $\Delta_0 := J(\boldsymbol{\theta}_0; \boldsymbol{\theta}_0) - \ell_{max}$. $\qquad\square$

# F   Additional Numerical Results

This section provides additional details for the numerical experiments that were omitted due to space limitation.

**Synthetic Data with Linear Model**   Fig. 3 shows the trajectories of training and testing accuracy with the SGD-GD scheme under different shift parameters, using the same settings as in Fig. 1 (left & right). Note that the testing dataset with 200 samples is generated from the same procedure described in the main paper with the same ground truth $\theta^o$, but without the label flipping step. In this case, although increasing the shift leads to a larger risk value $J(\theta_t; \theta_t)$ and more biased stationary solution in terms of $\|\nabla J(\theta_t; \theta_t)\|^2$ [cf. Fig. 1 (left & middle)], the test/train accuracy remain relatively stable regardless of the shift parameter. We remark that as observed from [Miller et al., 2021, Fig. 2], increasing the shift parameter $\epsilon$ does not always lead to a deteriorated or improved model accuracy. Importantly, the effects can be unpredictable in general, especially when only biased SPS solutions are guaranteed.

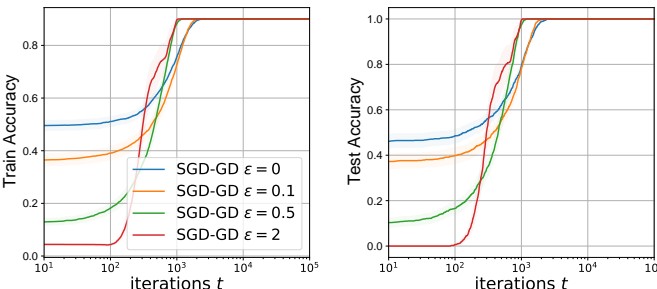

Figure 3: **Synthetic Data** (*Left*) Training accuracy under different sensitivity parameter $\epsilon_L$. (*Right*) Testing accuracy under different $\epsilon_L$.

Fig. 4 shows the trajectories of loss values $J(\theta_t; \theta_t)$, training and testing accuracy with the greedy and lazy deployment scheme using the same settings as in Fig. 1 (right). We observe similar behaviors as indicated in Fig. 3. Moreover, we notice that although the lazy deployment scheme converges to a less biased SPS solution than greedy deployment scheme utilizing the same number of samples, the initial convergence speed is slower. This can be predicted from Theorem 2 as the lazy deployment scheme is simulated with a larger noise variance $\sigma_0$.

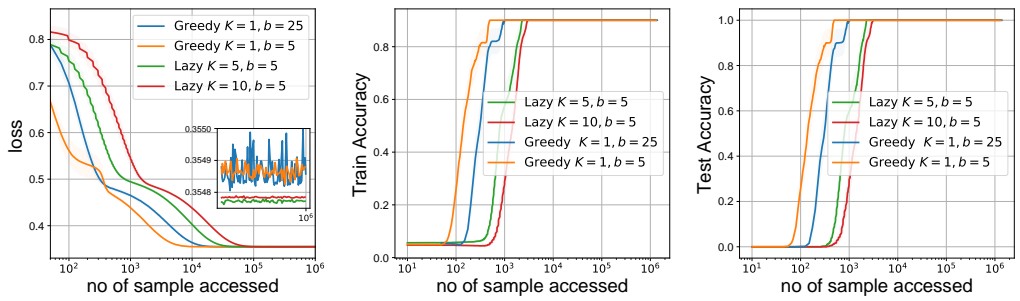

Figure 4: **Synthetic Data** (*Left*) Loss $V(\theta)$ against no of sample accessed. (*Middle*) Training accuracy under different sensitivity parameter $\epsilon_L$. (*Right*) Testing accuracy under different $\epsilon_L$.

**Real Data with Neural Network Model**   Similar to the above paragraph, Fig. 5, 6, 7 show the trajectories of train/test accuracy, for greedy/lazy deployment scheme when $\epsilon \in \{10, 10^4\}$ for completeness. The figures demonstrate similar behavior as described in the main paper. Moreover, we observe that the sensitivity parameter $\epsilon_{NN}$ has a small effect in the training/testing acccuracies of the trained models.

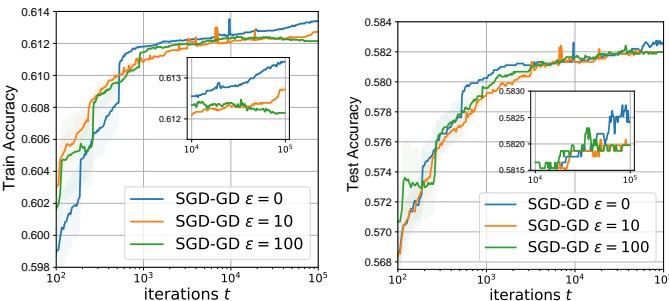

Figure 5: **Real Data with Neural Network** (*Left*) Training accuracy under different sensitivity parameter $\epsilon$. (*Right*) Testing accuracy under different $\epsilon$.

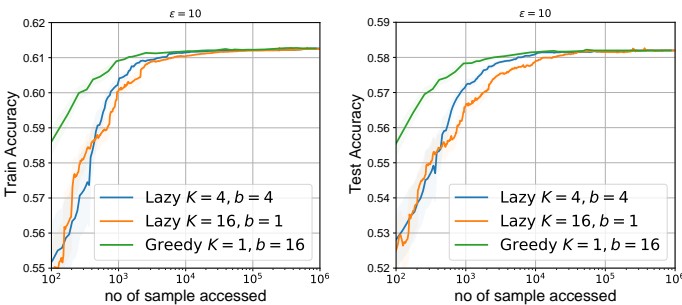

Figure 6: **Real Data with Neural Network** (*left & right*) Training accuracy under different deployment scheme when $\epsilon_{\mathsf{NN}} = 10$.

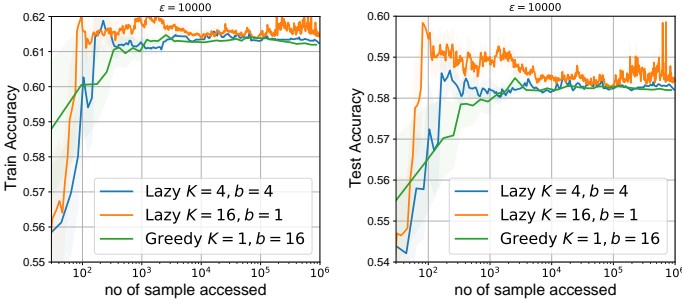

Figure 7: **Real Data with Neural Network** (*left & right*) Training accuracy under different deployment scheme when $\epsilon_{\mathsf{NN}} = 10^4$.

