# OpenReview forum: "Stochastic Optimization Schemes for Performative Prediction with Nonconvex Loss"
_NeurIPS.cc/2024/Conference — NeurIPS 2024 poster_

### Official Review · Reviewer_kiFq · 2024-07-08

**Soundness:** 4
**Presentation:** 4
**Contribution:** 3
**Rating:** 6
**Confidence:** 4

**Summary:**

A nice paper that studies nonconvex performative prediction optimization. Proposed a new stationarity notion and demonstrated convergence for SGD with greedy deployment.

**Strengths:**

1. Extending the convergence measurement from the strongly convex case to the nonconvex case and proposing the stationary performative stable notion.
2. Extremely clear and easy to follow with key insights for the performative prediction problems.
3. Novel convergence guarantees.
4. The lazy deployment is quite interesting. It is equivalent to using mini-batch in some sense.

**Weaknesses:**

Performative prediction problem is less motivated, i.e., there lacks a icon application such that the problem can only be formulated as a performative prediction problem and cannot be formulated in other forms even considering the special structure of the problem.
The numerical experiments lacks a convincing example as well, i.e., why studying the problem.

Leveraging the problem structure, many times the problem admits other more classical optimization objectives. It is not the problem of this paper alone but the whole research line.

**Questions:**

1. What happens to the analysis if the distribution is discrete. Then PDF may not exists and Pearson $\chi^2$ sensitivity may not be well-defined.
2. Regarding Theorem 2, to ensure $\delta$ stationarity, it requires $T=O(\delta^{-2})$ and $K=O(\delta^{-2})$. It means that to control both the bias and the error accumulated in the iterations, it needs $TK = O(\delta^{-4})$ iterations/samples. Is it possible to improve it to $O(\delta^{-2})$, i.e., the same complexity as the non performative setting?
3. Table 1 should reflect the bias level and compare the bias level with existing literature. It should also mention what would be the sample complexity needed to ensure an $\delta$ stationary point in this work and other works.

**Limitations:**

See questions above.

---

> ### Author Rebuttal · Authors · 2024-08-07
>
> > Performative prediction problem is less motivated, i.e., there lacks a icon application such that the problem can only be formulated as a performative prediction problem and cannot be formulated in other forms even considering the special structure of the problem. The numerical experiments lacks a convincing example as well, i.e., why studying the problem. Leveraging the problem structure, many times the problem admits other more classical optimization objectives. It is not the problem of this paper alone but the whole research line.
>
> Thanks for your critical comment on the state of performative prediction research. Many learning problems when applied in a "societal" setting will exhibit performativity. This is an inevitable outcome as the predictions informed by trained models becomes a part of the bigger social system. Further, an important property in performative prediction explored by this line of work is that the learner who trains the model cannot access information about how the distribution shifts, and thus cannot estimate the exact gradient of $V(\theta)$ that depends on ${\cal D}(\theta)$. Due to these limitations (note they are imposed by the application scenario rather than by the algorithm design), we believe that one of the key challenges shall lie in understanding and improving such "non-gradient" dynamics instead of exploring problem structure to efficiently minimize $V(\theta)$.
>
> > What happens to the analysis if the distribution is discrete. Then PDF may not exists and Pearson $\chi^2$ sensitivity may not be well-defined.
>
> That's a good observation. We agree that the chi-squared divergence condition in the original **C1** does not work with discrete distributions which limit its application. Fortunately, as pointed out by reviewer 71Gw, the above condition can be easily weakened into a sensitivity condition based on the total variation distance which also applies to discrete distribution (see **C1'** in the response to 71Gw). Nevertheless, our results also apply to cases with the Wasserstein-1 sensitivity condition (see **W1**).
>
>
> > Regarding Theorem 2, to ensure $\delta$ stationarity, it requires $T=O\left(\delta^{-2}\right)$ and $K=O\left(\delta^{-2}\right)$. It means that to control both the bias and the error accumulated in the iterations, it needs $T K=O\left(\delta^{-4}\right)$ iterations/samples. Is it possible to improve it to $O\left(\delta^{-2}\right)$, i.e., the same complexity as the non performative setting?
>
> We believe that it is possible to improve the sample complexity of $TK$ beyond $O(\delta^{-4})$. This is because from (22), we observe that within the $t$th "inner loop" after each deployment, the SGD-lazy deploy scheme is essentially SGD for $\min_{\theta} J(\theta;\theta_t)$. Now to improve the sample complexity with $K$, one may replace the SGD method with variance reduced SGD such as using the STORM gradient estimator in [a].
>
> However, we suspect that reaching the sample complexity of $O(\delta^{-2})$ would require more work. One of the reasons is that the structure of optimization algorithm is different as the learner does not have access to the (form of) distribution shift, and as explained in our manuscript, this limits the use of standard analysis tool such as (constant) Lyapunov function method. In general, characterizing and achieving the optimal sample complexity of performative optimization with non-convex loss is an exciting future direction to be explored.
>
> [a] Cutkosky and Orabona, "Momentum-based variance reduction in non-convex sgd", NeurIPS 2019.
>
> > Table 1 should reflect the bias level and compare the bias level with existing literature. It should also mention what would be the sample complexity needed to ensure an stationary point in this work and other works.
>
> The purpose of Table 1 is to compare the few existing works on non-convex performative prediction, yet other papers may have used other form of solution concepts where the studied algorithms admit various forms of biases. We intend to indicate their differences through displaying their respective convergent point "$\theta_{\infty}$" & algorithm type "Algo". To save space, we have used ${\cal O}(\epsilon)$-SPS to indicate the bias level of SGD-GD analyzed by us. As for the sample complexity, they can be deduced from the "Rate" column. We will improve the presentation of Table 1 in the revision if space allows.

---

> > ### Comment · Reviewer_kiFq · 2024-08-08
> > **Discussions**
> >
> > Thanks for the detailed responses. My concerns are mostly addressed. I will keep the score.
> >
> > Purely for discussions, understanding and improving "non-gradient" dynamics is definitely important. However, if in various applications of performative prediction optimization, there admit a more classical stochastic optimization formulation using additional structure that can decouple the source of randomness and the decision, it is unclear why one has to model it as a more general performative prediction optimization problem.

---

> > > ### Author Response · Authors · 2024-08-09
> > >
> > > Thank you for reading and replying to the responses.
> > >
> > > We agree with the reviewer that from the perspective of solving a stochastic optimization problem, using additional structure to decouple the source of randomness and decision may lead to better algorithms, e.g., lower sample complexity. However, we believe that an important aspect is that there are a number of scenarios for performative prediction where the learner does not **know** this "additional structure". Worse still, the learner may not even be aware of the distribution shift in the problem.
> > >
> > > A concrete example is the training of classifiers (e.g., for spam emails) - specifically in an online setting where the classifier has to be updated using current training data, note the latter may come from a decision dependent distribution $Z_{t+1} \sim D(\theta_t)$. On the other hand, it is likely that the learner does not know the form of decision-dependency in the training data since knowing the latter requires precisely knowing the behavior of the (normal and spam) email users. As a result, the only knowledge available to the learner is the current training data sample $Z_{t+1}$ and the form of cross entropy loss used for formulating the stochastic gradient $\nabla \ell( \theta; Z_{t+1} )$. In the absence of knowledge of $D(\theta)$, it would be impossible for the learner to exploit the problem structure and derive a reformulation to the performative optimization problem.

---

> > > > ### Comment · Reviewer_kiFq · 2024-08-09
> > > >
> > > > Thanks for the clarifications.

---

### Official Review · Reviewer_71Gw · 2024-07-11

**Soundness:** 4
**Presentation:** 3
**Contribution:** 3
**Rating:** 7
**Confidence:** 4

**Summary:**

The paper studies convergence of stochastic gradient descent in a performative prediction context. The main result shows that SGD converges to an analogue of performative stability, which the paper terms “stationary performative stability” (up to a bias term). The results characterize the rate of convergence and the magnitude of the bias.

**Strengths:**

The results in this paper significantly expand the scope of optimization in performative prediction, which has so far largely focused on convex loss functions. In fact, most results require strongly convex losses. Moreover, prior work typically makes an assumption on the magnitude of the performative effects, captured by the sensitivity parameter epsilon; the convergence results of this paper do not require a bound on epsilon and show that the sensitivity determines the distance to stationary performative stability. The lack of assumption about epsilon is a major advantage. There are some other works, e.g. Jagadeesan et al., that do not require a bound on epsilon, but this work requires knowing epsilon to run the optimization method. The additional analysis of the lazy deploy scheme, which approximates RRM and thus incurs no bias in the limit, is a nice addition. The observation about the different dependence of the bias on epsilon depending on whether the gradients are stochastic or not is another nice result.

**Weaknesses:**

This is not really a major weakness, but I think some of the discussion in Section 3.1 could be simplified. Instead of assuming the chi squared divergence condition, one can get Lemma 3 by assuming that D(theta) is Lipschitz in TV distance (i.e. ||D(\theta) - D(\theta')|| \leq \epsilon ||\theta - \theta'||), together with C2. The chi squared condition seems a bit odd and nonstandard because it is not a Lipschitz condition.

In A2, the first part of the sentence is not an assumption; it’s true just by the definition of J?

In Lemma 1, Theorem 1 (and possibly other places) please use parentheses in the step size condition. It should be 1/(L(1+sigma1^2)).

Very minor suggestion: personally I find it more appropriate to see Theorem 1 as a lemma and Corollary 1 as the main theorem.

Please don’t use the symbol T in Theorem 2 for the random step because you use that symbol in eq. (4).

**Questions:**

In the paragraph starting with line 275, you mention the relationship of Theorem 2 with RRM convergence and Mofakhami et al. My understanding was that they required a particular strong convexity condition, as noted in Table 1. So your result even for RRM may be new. Could you comment on this?

The discussion about the time-varying Lyapunov function (e.g. starting at line 179) reminded me of the perspective from Drusvyatskiy and Xiao. They show that SGD in a performative context can be thought of as standard SGD on the equilibrium distribution, at the stable point. I’m wondering if you’ve thought about if there exists an analogue of this perspective in your nonconvex setting?

**Limitations:**

There is not much discussion of limitations, though I don't think it is necessary.

---

> ### Author Rebuttal · Authors · 2024-08-07
>
> > I think some of the discussion in Section 3.1 could be simplified. Instead of assuming the chi squared divergence condition, one can get Lemma 3 by assuming that D(theta) is Lipschitz in TV distance (i.e. $||D(\theta) - D(\theta')|| \leq \epsilon ||\theta - \theta'||$), together with C2. The chi squared condition seems a bit odd and nonstandard because it is not a Lipschitz condition.
>
> Thanks for your valuable suggestion. Indeed, as you said, the chi-squared condition (**C1**) can be replaced by a weaker TV distance condition. The argument is as follows.
>
> First, we recall the definition of TV distance as $\delta_{TV}(\mu, \nu) = \sup_{A\subset {\sf Z} } \left| \mu(A) - \nu(A) \right| = \frac{1}{2} \int \left| p_{\mu}(z) - p_{\nu}(z) \right| {\sf d}z$ where $\mu, \nu$ are two measures supported on ${\sf Z}$ and $p_{(\cdot)}(z)$ denotes their pdfs. Note that we have $\delta_{TV}(\mu,\nu) \leq (1/2)\sqrt{\chi^2(\mu,\nu)}$ [Gibbs and Su, 2002, Sec. 2]. Accordingly, we may replace **C1** by its weakened version:
>
> **C1'** There exists a constant $\tilde{\epsilon}\geq 0$ such that $\delta_{TV}\left({\cal D}(\theta_1), {\cal D}(\theta_2)\right) \leq \tilde{\epsilon} \| \theta_1 - \theta_2 \|$ for any $\theta_1, \theta_2\in \mathbb{R}^d$
>
> Using **C1'** and **C2**, we derive a similar result as Lemma 3 with the following chain
> $$
> \begin{aligned}
>     \left| J(\theta, \theta_1) - J(\theta, \theta_2)\right| &= \left| \int \ell(\theta; Z) (p_{\theta_1}(z) - p_{\theta_2}(z)) {\sf d}(z) \right|
>     \leq \int |\ell(\theta; z)| \cdot \left| p_{\theta_1}(z) - p_{\theta_2}(z) \right| {\sf d}z
>     \leq \ell_{max} \cdot \int \left| p_{\theta_1}(z) - p_{\theta_2}(z) \right| {\sf d}(z)
> \end{aligned}
> $$
> $$\hspace{+3.5cm}
>  \leq \ell_{max} \cdot 2\delta_{TV}({\cal D}(\theta_1), {\cal D}(\theta_2))
>     \leq 2\ell_{max} \tilde{\epsilon} || \theta_1 - \theta_2 ||
> $$
> The rest of our convergence analysis for SGD-GD or SGD-lazy deploy follows immediately with the above modification. Again, we thank the reviewer for pointing this out and will make sure to include the above proofs in the revision!
>
> > Personally, I find it more appropriate to see Theorem 1 as a lemma and Corollary 1 as the main theorem.
>
> We chose to describe our theoretical results in this way as we wish to include general conditions for the step size $\gamma_{t}$, such as diminishing and constant step sizes.
>
> > In the paragraph starting with line 275, you mention the relationship of Theorem 2 with RRM convergence and Mofakhami et al. My understanding was that they required a particular strong convexity condition, as noted in Table 1. So your result even for RRM may be new. Could you comment on this?
>
> Though our Theorem 2 suggests a new finding for an RRM-like strategy, we believe that such strategy of SGD + lazy deployment with $K \to \infty$ strategy is not strictly equivalent to RRM. A subtle difference is that RRM (e.g., in [Mofakhami et al.]) requires finding an exact minimizer to the risk minimization problem given a fixed data distribution at each iteration, yet SGD + lazy deployment may only find a stationary point to the risk minimization problem (with non-convex loss) even when $K \to \infty$. As such, we have restrained from claiming it as a new finding for RRM. We will elaborate more in the revision.
>
>
> > The discussion about the time-varying Lyapunov function (e.g. starting at line 179) reminded me of the perspective from Drusvyatskiy and Xiao. They show that SGD in a performative context can be thought of as standard SGD on the equilibrium distribution, at the stable point. I’m wondering if you’ve thought about if there exists an analogue of this perspective in your nonconvex setting?
>
> Although this is an interesting point, we believe that drawing such analogue for the nonconvex setting is difficult since the equilibrium distribution may not be unique in the latter case. In fact, the inability to apply the Lyapunov function $J(\theta;\theta_{PS})$ is also the reason why we had to develop a new time varying Lyapunov function in the nonconvex setting.
>
> > Other Problems
>
> We also thank the reviewer for pointing out other typos and small issues in the paper. We will correct them in the revision.

---

> > ### Comment · Reviewer_71Gw · 2024-08-10
> >
> > Thank you for the detailed response! Everything makes sense.

---

### Official Review · Reviewer_JTrB · 2024-07-26

**Soundness:** 3
**Presentation:** 3
**Contribution:** 3
**Rating:** 7
**Confidence:** 4

**Summary:**

This paper studied the `performative prediction’ that means when predictive models are used to make consequential decisions like policy making, it can trigger actions that influence the outcome they aim to predict. And we know a system with unlimited positive feedback will eventually be destroyed. On the optimization side, this work considered a risk minimization problem with a decision-dependent data distribution. It means the loss function specified by model parameters $\theta$ and the data distribution. They analyzed stochastic gradient descent (SGD) with a greedy deployment scheme (SGD-GD) in a setting that only requires the smoothness of non-convex loss function $l$. They showed the algorithm convergence to stationary performative stable (SPS) solutions with two types of distance metrics of distributions. Numerical examples of both synthetic and real data are provided to justify their theoretical result.

**Strengths:**

Pros:

It's a solid extension of the SGD-GD works of [Mofakhami et al., 2023] and
[Mendler-Dünner et al., 2020] with smooth but not necessarily non-convex loss $l$. It’s a big step forward compared to the strongly convex loss in previous work.

They provide both real and synthetic experiments to justify their claims.

**Weaknesses:**

Cons:

The experiment setting is relatively simple but it's a minor issue since this is a theoretical work and the experiment is showcasing the concept.

**Questions:**

The reviewer is generally positive about this work. One question would be:

Are there some other real applications besides the spam filter? Could the author provide more insight into this theoretical work and other policy-making applications with real-world influence?

---

> ### Author Rebuttal · Authors · 2024-08-07
>
> > The experiment setting is relatively simple but it's a minor issue since this is a theoretical work and the experiment is showcasing the concept.
>
> We chose a simple experiment setting to demonstrate the effects of key parameters such as sensitivity strength $\epsilon$ and lazy deployment period $K$ to better validate the theoretical findings. We believe that this is an appropriate choice given the theoretical nature of this work.
>
>
> > Are there some other real applications besides the spam filter? Could the author provide more insight into this theoretical work and other policy-making applications with real-world influence?
>
> Examples of performativity are pervasive in the real world, especially in the financial market and strategic training tasks, such as insurance, hiring, admission, and healthcare. In these scenarios, individuals often adjust their behaviors to receive predictions in their favor. For instance, in the hiring process, job applicants may prepare more relevant projects based on the job description provided by a company’s HR. When an employer conducts an interview and decides whether to hire an applicant, the applicant may have a higher chance of being hired if s/he is better prepared.
>
> From a theoretical perspective, a crucial point we explored in this paper is that decisions are often made via models that are trained thru a non-convex optimization process, which has not been addressed in previous works on performative prediction. One takeaway from our findings is that - with reference to the results on lazy-deployment vs greedy-deployment SGD - when a decision maker (company) frequently changes its job description requirements, its trained model may experience greater bias that may lead to reduced performance.

---

> > ### Comment · Reviewer_JTrB · 2024-08-13
> > **Response to the author**
> >
> > The reviewer thanks the author for the response. After reading the rebuttal discussion from all reviewers, the reviewer would like to maintain the score.

---

### Official Review · Reviewer_eMX9 · 2024-07-29

**Soundness:** 3
**Presentation:** 3
**Contribution:** 3
**Rating:** 6
**Confidence:** 4

**Summary:**

This work studied performative prediction problems in nonconvex regimes and proposed the first algorithm, SGD-GD, with convergence guarantees in this case, it was further extended to a lazy deployment scheme so that the algorithm is bias-free.

**Strengths:**

1. First convergence analysis of gradient-based algorithms for performative prediction problems in nonconvex regimes. Which is a novel contribution.
2. Proposed a new convergence measurement for nonconvex performative prediction problems
3. The writing is great and the storyline is easy to understand

**Weaknesses:**

1. The definition of SPS, as the authors mentioned, only considers the gradient regarding the loss function, while missing the gradient over the distribution parameter, which may not perfectly reflect the stationarity convergence of the objective function.
2. Some assumptions are still a bit unrealistic (for example, the global upper bound assumption in C2), it is not clear whether they are satisfied in the numerical experiments.

**Questions:**

/

**Limitations:**

/

---

> ### Author Rebuttal · Authors · 2024-08-07
>
> > The definition of SPS, as the authors mentioned, only considers the gradient regarding the loss function, while missing the gradient over the distribution parameter, which may not perfectly reflect the stationarity convergence of the objective function.
>
> This is a valid observation. However, we remark that SPS is a stationary solution concept suitable for the arguably more "natural" algorithms for performative prediction -- including SGD-GD, SGD with lazy deployment, repeated risk minimization. These algorithms do not require a-priori knowledge of the form of data distribution shift, nor attempt to learn the latter. We believe these class of algorithms are important as they are applicable to scenarios when the learner is agnostic to the distribution shift. The definition of SPS generalizes that of performative stable (PS) solution [Perdomo et al., 2020] since for (strongly) convex $\ell(.)$, the definitions of SPS and PS solutions are equivalent. Note algorithms for achieving a stationary point for $V(\theta)$ has been explored in works like [Izzo et al., 2021], which represents a different research direction in the performative prediction community.
>
>
> > Some assumptions are still a bit unrealistic (for example, the global upper bound assumption in C2), it is not clear whether they are satisfied in the numerical experiments.
>
> Admittedly, while assumptions like C2 may appear to be slightly strong, our theory remains applicable to a number of applications in ML including our numerical experiments. For the synthetic data experiments, the sigmoid loss function is upper bounded by 1. Similarly, in the neural network experiments, the loss is also upper bounded due to the sigmoid function applied in the last layer. Both simulations satisfy one of the required sets of assumptions (W1+W2 or C1+C2). We will expand the discussion on how these examples satisfy the assumptions in the revision. We also remark that C1 can be further relaxed to **C1'** using a weakened notion of distribution sensitivity; see the response to 71Gw.

---

### Author Rebuttal · Authors · 2024-08-07

### General Response
We thank all the four reviewers for their careful reviews and valuable suggestions. We summarize our general responses and proposed improvement to the paper as follows:

- Our work provides one of the first convergence theories on SGD-greedy deployment scheme applied to performative prediction with non-convex losses, i.e., a stochastic optimization problem with decision dependent samples. To do so, we developed several innovations: (a) the concept of SPS solution for "equilibrium" solution defined w.r.t. a partial stationarity condition, (b) a time varying Lyapunov function that tracks the progress of SGD-GD in the absence of a unique equilibrium solution. These innovations have led to new findings that seems to be unique for non-convex performative prediction, namely, SGD-GD may converge to a biased SPS solution, and the bias can be reduced/eliminated with a lazy deployment variant. We believe that these findings/innovations will be of interest to the performative prediction community, as well as studies on stochastic algorithms in general.
- We are grateful for the constructive suggestions. Particularly, we have weakened the condition **C1** based on $\chi^2$ divergence into that based on the weaker TV distance; see **C1'** in the response to reviewer 71Gw. This allows our main theories to be applied on more general settings for performative prediction.
- We also thank reviewer kiFq for raising the issue about reducing the sample complexity to reach an (unbiased) SPS solution. We listed some ideas in the response below but this is an important future direction that we would like to explore.

We found that these suggestions improves the paper quality and will include them into the revision. We look forward to further discussion with the reviewers in the next phase of this review process. Thank you!

---

### Decision · Program_Chairs · 2024-09-25

**Decision:**

Accept (poster)

**Comment:**

This paper considers a variant of SGD where the distributions of stochastic gradients depend on the values of $\theta_t$ when applied to nonconvex problems. Due to this variation, the original proof of SGD converges to a stationary point cannot be easily applied. The main contribution of this paper is designing the decoupled performative risk $J$ and the decoupled partial gradient $\partial J$ in (3). Then the authors manage to apply similar steps of plain SGD except that the sum of the norms of full gradients are bounded by the $J$ functions, not the batch objective $V$, which is quite clever.